# KERNELWAREHOUSE: RETHINKING THE DESIGN OF DYNAMIC CONVOLUTION

## ABSTRACT

Dynamic convolution learns a linear mixture of $n$ static kernels weighted with their sample-dependent attentions, demonstrating superior performance than normal convolution. However, it increases the number of convolutional parameters by $n$ times. This and the optimization difficulty lead to no research progress that can allow researchers to use a significant large value of $n$ (e.g., $n > 100$ instead of typical setting $n < 10$) to push forward the performance boundary of dynamic convolution while enjoying parameter efficiency. To fill this gap, in this paper, we rethink the basic concepts of "**kernels**", "**assembling kernels**" and "**attention function**" in dynamic convolution through the lens of exploiting convolutional parameter dependencies within the same layer and across successive layers, and propose **KernelWarehouse**. As a more general form of dynamic convolution, KernelWarehouse provides a high degree of freedom to fit a desired parameter budget under large kernel numbers (e.g., $n = 108$ or $n = 188$). We compare our method and existing dynamic convolution methods on ImageNet and MS-COCO datasets using various ConvNet architectures, and show that it attains state-of-the-art results. For instance, the ResNet18|ResNet50|MobileNetV2|ConvNeXt-Tiny model trained with KernelWarehouse on ImageNet reaches 76.05%|81.05%|75.92%|82.55% top-1 accuracy. Thanks to its flexible design, KernelWarehouse can even reduce the model size of a ConvNet while improving the accuracy, e.g., our ResNet18 model with 36.45%|65.10% parameter reduction to the baseline shows 2.89%|2.29% absolute improvement to top-1 accuracy. Code is provided for results reproduction.

## 1 INTRODUCTION

Convolution is the key operation in convolutional neural networks (ConvNets) (Krizhevsky et al., 2012; Szegedy et al., 2015; He et al., 2016; Howard et al., 2017; Tan & Le, 2019a; Liu et al., 2022). In a convolutional layer, normal convolution $\mathbf{y} = \mathbf{W} * \mathbf{x}$ computes the output $\mathbf{y}$ by applying the same convolutional kernel $\mathbf{W}$ defined as a set of convolutional filters to every input sample $\mathbf{x}$. *For brevity, we refer to "convolutional kernel" as "kernel" and omit the bias term throughout this paper.* Although the efficacy of normal convolution is extensively validated with various types of ConvNets on many computer vision tasks, recent progress in the efficient ConvNet architecture design shows that dynamic convolution, also known as CondConv (Yang et al., 2019a) and DY-Conv (Chen et al., 2020), achieves large performance gains. The basic idea of dynamic convolution is to replace the static kernel in normal convolution by a linear mixture of $n$ same dimensioned kernels $\mathbf{W} = \alpha_1 \mathbf{W}_1 + ... + \alpha_n \mathbf{W}_n$, where $\alpha_1, ..., \alpha_n$ are scalar attentions generated by an input-dependent attention module. Benefiting from the additive property of $n$ kernels $\mathbf{W}_1, ..., \mathbf{W}_n$ and compact attention module designs, dynamic convolution improves the feature learning ability with little extra multiply-add cost. However, it increases the number of convolutional parameters by $n$ times, which leads to a huge rise in model size because the convolutional layers of a modern ConvNet occupy the vast majority of parameters.

DCD (Li et al., 2021b) learns a base kernel and a sparse residual to approximate dynamic convolution via matrix decomposition. This coarse approximation abandons the basic mixture learning paradigm, and cannot well retain the representation power of dynamic convolution when $n$ becomes large. ODConv (Li et al., 2022) presents a more powerful attention module to dynamically weight static kernels along different dimensions instead of one single dimension, which can get competitive performance with a reduced number of kernels. However, under the same setting of $n$, ODConv has more parameters than vanilla dynamic convolution. He et al. (2023) directly uses popular weight

pruning strategy to compress DY-Conv via multiple pruning-and-retraining phases. In brief, existing dynamic convolution methods based on the mixture learning paradigm are not parameter-efficient. Restricted by this, they typically set $n = 8$ (Yang et al., 2019a) or $n = 4$ (Chen et al., 2020; Li et al., 2022). More importantly, a plain fact is that the improved capacity of a ConvNet constructed with dynamic convolution comes from increasing the number of kernels per convolutional layer facilitated by the attention mechanism. *This causes an intrinsic conflict between the desired model size and capacity, which prevents researchers to explore the performance boundary of dynamic convolution with a significantly large kernel number (e.g., $n > 100$) while enjoying parameter efficiency.* In this paper, we attempt to break down this barrier by rethinking the design of dynamic convolution.

Specifically, we present *KernelWarehouse* (see Figure 1 for a schematic overview), a more general form of dynamic convolution, which can strike a favorable trade-off between parameter efficiency and representation power. The formulation of KernelWarehouse is inspired by two observations. On the one hand, we note that existing dynamic convolution methods treat all parameters in a regular convolutional layer as a kernel and increase the kernel number from 1 to $n$, then use their attention modules to assemble $n$ kernels into a linearly mixed kernel. Though straightforward and effective, they pay no attention to parameter dependencies within the static kernel at a convolutional layer. On the other hand, we notice that existing dynamic convolution methods allocate a different set of $n$ kernels for every individual convolutional layer of a ConvNet, ignoring parameter dependencies across successive layers. We hypothesize that the barrier can be removed by way of flexibly exploiting these two types of convolutional parameter dependencies for redefining dynamic convolution.

Driven by the above analysis, we introduce two simple strategies to formulate KernelWarehouse, namely kernel partition and warehouse sharing. With kernel partition, the static kernel for a regular convolutional layer is sequentially divided into $m$ disjoint kernel cells of the same dimensions along spatial and channel dimensions, and then the linear mixture can be defined in terms of much smaller local kernel cells instead of holistic kernels. Specifically, each of $m$ kernel cells is represented as a linear mixture based on a predefined "*warehouse*" consisting of $n$ kernel cells (e.g., $n = 108$), and the static kernel will be replaced by assembling its corresponding $m$ mixtures in order. With warehouse sharing, multiple neighboring convolutional layers of a ConvNet can share the same warehouse as long as the same kernel cell size is used in the kernel partition process, further enhancing its parameter efficiency and representation power to use a larger value of $n$ given a desired parameter budget. Nevertheless, with a significantly large value of $n$, the optimization of KernelWarehouse becomes fundamentally more challenging compared to existing dynamic convolution methods, making popular attention functions used in dynamic convolution research lose effectiveness under this circumstance. We solve this problem by defining a contrasting-driven attention function (CAF). By these components, we redefine the basic concepts of "*kernels*", "*assembling kernels*" and "*attention function*" in dynamic convolution from the perspective of reducing kernel dimension and increasing kernel number significantly, taking advantage of the aforementioned parameter dependencies as easy as possible. These simple concept shifts provide a high degree of flexibility for KernelWarehouse to balance parameter efficiency and representation power under different parameter budgets.

As a drop-in replacement of normal convolution, KernelWarehouse can be easily used to different ConvNet architectures. Extensive experiments on ImageNet and MS-COCO datasets show that our method achieves better results than its dynamic convolution counterparts.

## 2  RELATED WORK

**ConvNet Architectures.** In the past decade, a lot of top-performing ConvNet architectures such as AlexNet (Krizhevsky et al., 2012), VGGNet (Simonyan & Zisserman, 2015), GoogLeNet (Szegedy et al., 2015), ResNet (He et al., 2016), DenseNet (Huang et al., 2017), ResNeXt (Xie et al., 2017) and RegNet (Radosavovic et al., 2020) have been presented. Around the same time, some lightweight architectures like MobileNet (Howard et al., 2017; Sandler et al., 2018; Howard et al., 2019), ShuffleNet (Zhang et al., 2018b) and EfficientNet (Tan & Le, 2019a) have been designed for resource-constrained applications. Recently, Liu et al. (2022) presented ConvNeXt whose performance can match newly emerging vision transformers (Dosovitskiy et al., 2021; Liu et al., 2021). As a plug-and-play design, our method could be used to improve their performance.

**Feature Recalibration.** An effective way to enhance the capacity of a ConvNet is feature recalibration. It relies on attention mechanisms to adaptively refine the feature maps learnt by a convolutional

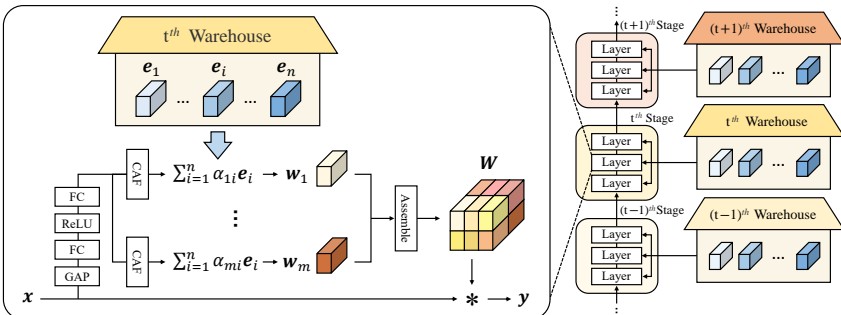

Figure 1: A schematic illustration of KernelWarehouse. Please see the Method section for details.

block. Popular feature recalibration modules such as RAN (Wang et al., 2017), SE (Hu et al., 2018b), BAM (Park et al., 2018), CBAM (Woo et al., 2018), GE (Hu et al., 2018a), SRM (Lee et al., 2019) and ECA (Wang et al., 2020) focus on different design aspects: using channel attention, or spatial attention, or hybrid attention to emphasize important features and suppress unnecessary ones. Unlike these methods which retain the static kernel, dynamic convolution replaces the static kernel of a convolutional layer by a linear mixture of $n$ kernels weighted with the attention mechanism.

**Dynamic Weight Networks.** Many research efforts have been made on developing effective methods to generate the weights for a neural network. Jaderberg et al. (2015) proposes a Spatial Transformer module which uses a localisation network that predicts the feature transformation parameters conditioned on the learnt feature itself. Dynamic Filter Network (Jia et al., 2016) and Kernel Prediction Networks (Bako et al., 2017; Mildenhall et al., 2018) introduce two filter generation frameworks which share the same idea: using a deep neural network to generate sample-adaptive filters conditioned on the input. Based on this idea, DynamoNet (Diba et al., 2019) uses dynamically generated motion filters to boost human action recognition in videos. CARAFE (Wang et al., 2019) and Involution (Li et al., 2021a) further extend this idea by designing efficient generation modules to predict the weights for extracting informative features. By connecting this idea with SE, WeightNet (Ma et al., 2020), CGC (Lin et al., 2020) and WE (Quader et al., 2020) design different attention modules to adjust the weights in convolutional layers of a ConvNet. Hypernetwork (Ha et al., 2017) uses a small network to generate the weights for a larger recurrent network instead of a ConvNet. MetaNet (Munkhdalai & Yu, 2017) introduces a meta learning model consisting of a base learner and a meta learner, allowing the learnt network for rapid generalization across different tasks. This paper focuses on advancing dynamic convolution research, which differs from these works in formulation.

## 3 METHOD

In this section, we describe the formulation of KernelWarehouse and clarify its key components.

### 3.1 FORMULATION OF KERNELWAREHOUSE

For a convolutional layer, let $\mathbf{x} \in \mathbb{R}^{h \times w \times c}$ and $\mathbf{y} \in \mathbb{R}^{h \times w \times f}$ be the input having $c$ feature channels with the resolution $h \times w$ and the output having $f$ feature channels with the same resolution to the input, respectively. Normal convolution $\mathbf{y} = \mathbf{W} * \mathbf{x}$ uses a single static kernel $\mathbf{W} \in \mathbb{R}^{k \times k \times c \times f}$ consisting of $f$ convolutional filters with the spatial size $k \times k$, while dynamic convolution replaces the static kernel by $\mathbf{W} = \alpha_1 \mathbf{W}_1 + ... + \alpha_n \mathbf{W}_n$, a linear mixture of $n$ same dimensioned static kernels weighted with their input-dependent scalar attentions $\alpha_1, ..., \alpha_n$. In sharp contrast to existing methods (Yang et al., 2019a; Chen et al., 2020; Li et al., 2022), our KernelWarehouse redefines the basic concepts of "*kernels*", "*assembling kernels*" and "*attention function*" by applying the attentive mixture learning paradigm to a dense local kernel scale instead of a holistic kernel scale via three coupled components: kernel partition, warehouse sharing and contrasting-driven attention function.

**Kernel Partition.** The basic idea of kernel partition is to reduce kernel dimension and increase kernel number via explicitly exploiting parameter dependencies within the same convolutional layer. Firstly, we sequentially divide the static kernel $\mathbf{W}$ at a regular convolutional layer along spatial and channel dimensions into $m$ disjoint parts $\mathbf{w}_1, ..., \mathbf{w}_m$ called "*kernel cells*" that have the same dimensions. *For brevity, here we omit to define kernel cell dimensions.* Kernel partition can be defined as

$$\mathbf{W} = \mathbf{w}_1 \cup ... \cup \mathbf{w}_m, \text{ and } \forall \, i, j \in \{1, ..., m\}, i \neq j, \; \mathbf{w}_i \cap \mathbf{w}_j = \emptyset. \tag{1}$$

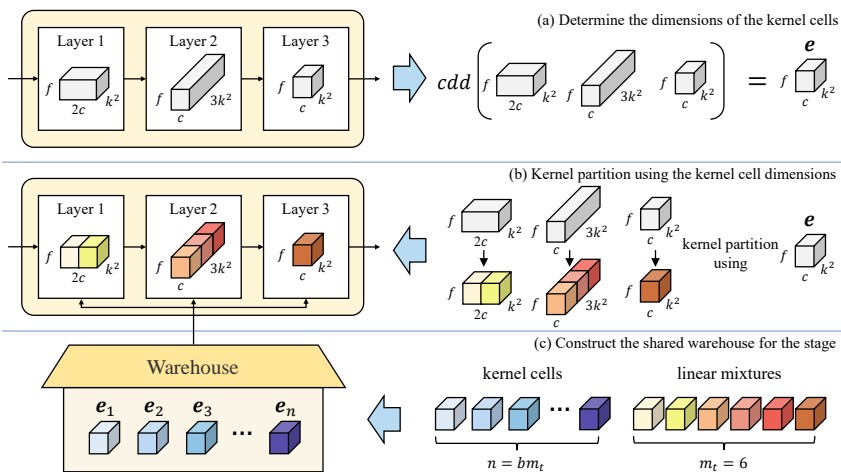

Figure 2: An illustration of kernel partition, warehouse construction and sharing for KernelWarehouse in the same-stage convolutional layers of a ConvNet. $cdd$ denotes common kernel dimension divisors.

After kernel partition, we treat kernel cells $\mathbf{w}_1,...,\mathbf{w}_m$ as "*local kernels*", and define a "*warehouse*" containing $n$ kernel cells $\mathbf{E} = \{\mathbf{e}_1, ..., \mathbf{e}_n\}$, where $\mathbf{e}_1, ..., \mathbf{e}_n$ have the same dimensions as $\mathbf{w}_1,...,\mathbf{w}_m$. Then, by sharing the warehouse $\mathbf{E} = \{\mathbf{e}_1, ..., \mathbf{e}_n\}$ in the same convolutional layer, we can now represent each of $m$ kernel cells $\mathbf{w}_1,...,\mathbf{w}_m$ as

$$\mathbf{w}_i = \alpha_{i1}\mathbf{e}_1 + ... + \alpha_{in}\mathbf{e}_n, \text{ and } i \in \{1, ..., m\}, \tag{2}$$

where $\alpha_{i1}, ..., \alpha_{in}$ are the scalar attentions generated by an attention module $\phi(x)$ conditioned on the input $\mathbf{x}$. Finally, the static kernel $\mathbf{W}$ in a regular convolutional layer is replaced by assembling its corresponding $m$ linear mixtures in order. Note that the dimensions of a kernel cell can be much smaller than the dimensions of the static kernel $\mathbf{W}$ (e.g., when $m = 16$, the number of parameters in a kernel cell is 1/16 to that of the static kernel $\mathbf{W}$). Under the same parameter budget, this allows a warehouse can have a large setting of $n$ (e.g., $n > 100$), in sharp contrast to existing dynamic convolution methods which define the linear mixture in terms of $n$ "*holistic kernels*" and typically use a much smaller setting of $n$ (e.g., $n < 10$) restricted by their parameter-inefficient shortcoming.

**Warehouse Sharing.** The main goal of warehouse sharing is to further improve parameter efficiency and representation power of KernelWarehouse through explicitly exploiting parameter dependencies across successive convolutional layers. Specifically, we share a warehouse $\mathbf{E} = \{\mathbf{e}_1, ..., \mathbf{e}_n\}$ to represent every kernel cell at $l$ neighboring convolutional layers in the same-stage building blocks of a ConvNet, allowing it can use a much larger setting of $n$ against kernel partition under the same parameter budget. This is mostly easy, as modern ConvNets such as ResNet (He et al., 2016), MobileNet (Howard et al., 2017) and ConvNeXt (Liu et al., 2022) typically adopt a modular design scheme in which static kernels in the same-stage convolutional layers usually have the same or regular dimensions. In implementation, given a desired convolutional parameter budget (which will be defined later), we simply use common dimension divisors for $l$ static kernels as the uniform kernel cell dimensions for kernel partition, which naturally determine the values of $m$ and $n$. Besides, parameter dependencies across neighboring convolutional layers are also critical in strengthening the capacity of a ConvNet (see Table 6). Figure 2 illustrates how kernel partition, warehouse construction and sharing in the same-stage convolutional layers of a ConvNet are done.

**Parameter Efficiency and Representation Power.** Let $n$ be the number of kernel cells in a warehouse shared to $l$ convolutional layers of a ConvNet, and let $m_t$ be the total number of kernel cells in these $l$ convolutional layers ($m_t = m$, when $l = 1$). Then, **we define** $b = n/m_t$ as a scaling factor to indicate the convolutional parameter budget of KernelWarehouse relative to normal convolution. Here, we do not consider the number of parameters in the attention module $\phi(x)$ which generates $nm_t$ scalar attentions, as it is much smaller than the total number of parameters for normal convolution at $l$ convolutional layers. In implementation, we use the same value of $b$ to all convolutional layers of every ConvNet. Under this condition, we can see that KernelWarehouse can easily scale up or scale down the model size of a ConvNet by changing $b$. Compared to normal convolution: (1) when $b < 1$, KernelWarehouse tends to get the reduced model size; (2) when $b = 1$, KernelWarehouse tends to get the similar model size; (3) when $b > 1$, KernelWarehouse tends to get the increased model size.

---

**Algorithm 1:** Implementation of KernelWarehouse

**Part-A: Kernel Partition and Warehouse Construction**

**Require:** network $\mathbf{M}$ consisting of $S$ convolutional stages, parameter budget $b$

1 **for** $s \leftarrow 1$ **to** $S$ **do**
2     $\{\mathbf{W}_i \in \mathbb{R}^{k_i \times k_i \times c_i \times f_i}\}_{i=1}^l$ = static kernels in the $s$-th stage of $\mathbf{M}$
3     $k_e, c_e, f_e \leftarrow \mathbf{cdd}(\{k_i\}_{i=1}^l), \mathbf{cdd}(\{c_i\}_{i=1}^l), \mathbf{cdd}(\{f_i\}_{i=1}^l)$
4     $m_t \leftarrow 0$
5     **for** $i \leftarrow 1$ **to** $l$ **do**
6        $\{\mathbf{w}_j \in \mathbb{R}^{k_e \times k_e \times c_e \times f_e}\}_{j=1}^m \leftarrow \mathbf{kernel\_partition}(\mathbf{W}_i, k_e, c_e, f_e)$
7        $\mathbf{W}_i \leftarrow \mathbf{w}_1 \cup \cdots \cup \mathbf{w}_m$, and $\forall i, j \in \{1, \ldots, m\}, i \neq j, w_i \cap w_j = \varnothing$
8        $m \leftarrow k_i k_i c_i f_i / k_e k_e c_e f_e$
9        $m_t \leftarrow m + m_t$
10     **end**
11     $n \leftarrow b m_t$
12     $\mathbf{E}_s \leftarrow \{\mathbf{e}_i \in \mathbb{R}^{k_e \times k_e \times c_e \times f_e}\}_{i=1}^n$
13 **end**
14 $\mathbf{E} \leftarrow \{\mathbf{E}_1, \ldots, \mathbf{E}_S\}$

**Return :** network $\mathbf{M}$ with partitioned kernels, set $\mathbf{E}$ consisting of $S$ warehouses

---

**Part-B: Kernel Assembling for Single Same-Stage Convolutional Layer**

**Require:** input $\mathbf{x}$, attention module $\phi$, warehouse $\mathbf{E} = \{\mathbf{e}_i\}_{i=1}^n$, linear mixtures $\{\mathbf{w}_i\}_{i=1}^m$

1 $\alpha \leftarrow \phi(\mathbf{x})$
2 **for** $i \leftarrow 1$ **to** $m$ **do**
3     $\mathbf{w_i} \leftarrow \alpha_{i1} \mathbf{e}_1 + \cdots + \alpha_{in} \mathbf{e}_n$
4 **end**
5 $\mathbf{W} \leftarrow \mathbf{w}_1 \cup \cdots \cup \mathbf{w}_m$, and $\forall i, j \in \{1, \ldots, m\}, i \neq j, w_i \cap w_j = \varnothing$

**Return :** assembled kernel $\mathbf{W}$

---

Intriguingly, given a desired value of $b$, a proper and large value of $n$ can be obtained by simply changing $m_t$, providing a representation power guarantee for KernelWarehouse. Therefore, Kernel-Warehouse can strike a favorable trade-off between parameter efficiency and representation power, under different convolutional parameter budgets of $b$, as illustrated by trained model exemplifications KW ($1/2\times, 3/4\times, 1\times, 4\times$) in Table 1 and Table 2. Algorithm 1 shows the implementation of KernelWarehouse, given a ConvNet backbone and the expired convolutional parameter budget $b$.

**Discussion.** It should be noted that the split-and-merge mechanism with multi-branch group convolution has been widely used in many existing works (Szegedy et al., 2015; Xie et al., 2017; Sandler et al., 2018; Li et al., 2019; Tan & Le, 2019b; Yang et al., 2019b; Tan & Le, 2019b; Li et al., 2020; Liu et al., 2022) to enhance the capacity of ConvNets. Although KernelWarehouse also uses the parameter splitting idea in the kernel partition component, its focus and motivation we have clarified above are clearly different from them. Besides, KernelWarehouse could be naturally used to improve their performance as they are based on normal convolution. We have validated this on MobileNetV2 and ConvNeXt (see Table 1 and Table 2).

According to the formulation of KernelWarehouse, it will degenerate into vanilla dynamic convolution (Yang et al., 2019a; Chen et al., 2020) when uniformly setting $m = 1$ in kernel partition (i.e., all kernel cells in each warehouse have the same dimensions as the static kernel $\mathbf{W}$ in normal convolution) and $l = 1$ in warehouse sharing (i.e., each warehouse is limited to its specific convolutional layer). Therefore, KernelWarehouse is a more general form of dynamic convolution.

### 3.2 ATTENTION FUNCTION OF KERNELWAREHOUSE

Designing a proper attention function is essential to the optimization of KernelWarehouse. The new formulation of KernelWarehouse brings three unique optimization properties: (1) the attentive mixture learning is applied to a dense local kernel cell scale instead of a holistic kernel scale; (2) the number of kernel cells in a warehouse is significantly large (e.g., $n > 100$ vs. $n < 10$); (3) a warehouse is not only shared to represent each of $m$ kernel cells for a specific convolutional layer of a ConvNet, but also is shared to represent every kernel cell for the other $l - 1$ same-stage convolutional layers. Under this learning context, we empirically find that popular attention functions such as Sigmoid and Softmax lose effectiveness for KernelWarehouse. Even with the temperature annealing (Li et al., 2022), they get worse results than vanilla dynamic convolution (see Table 8).

**Contrasting-driven Attention Function.** We present a contrasting-driven attention function (CAF) which can well fit the optimization properties of KernelWarehouse. For $i^{th}$ kernel cell in the static

kernel $\mathbf{W}$, let $z_{i1}, ..., z_{in}$ be the feature logits generated by the second fully-connected layer of a compact SE-typed structure $\phi(x)$ (**see Appendix for its detailed structure**), then CAF is defined as

$$\alpha_{ij} = (1 - \tau)\frac{z_{ij}}{\sum_{p=1}^{n} |z_{ip}|} + \tau\beta_{ij}, \text{ and } j \in \{1, ..., n\}, \tag{3}$$

where $\tau$ is a temperature linearly reducing from 1 to 0 in the early training stage; $\frac{z_{ij}}{\sum_{p=1}^{n} |z_{ip}|}$ is a linear normalization function; $\beta_{ij}$ is a binary value (0 or 1) which is used for initializing the attentions.

In principle, (1) the second term of CAF ensures that the initial valid kernel cells ($\beta_{ij} = 1$) in a shared warehouse are uniformly allocated to represent different linear mixtures at the same-stage convolutional layers at the beginning of the model training; (2) the first term enables the existence of both negative and positive attentions (in contrast to popular attention functions that do not generate negative attentions) and encourages the optimization process to learn contrasting and diverse attention relationships among all linear mixtures sharing the same warehouse, making the mixed kernel cells at $l$ convolutional layers can learn informative and discriminative features hieratically. The setting of $\beta_{ij}$ at $l$ convolutional layers should assure the shared warehouse can assign: (1) at least one specified kernel cell ($\beta_{ij} = 1$) to every linear mixture, given a desired convolutional parameter budget $b \geq 1$; (2) at most one specific kernel cell ($\beta_{ij} = 1$) to every linear mixture, given $b < 1$. In implementation, we adopt a simple strategy to assign one of the total $n$ kernel cells in a shared warehouse to each of $m_t$ linear mixtures at $l$ convolutional layers without repetition. When $n$ is less than $m_t$, we let the remaining linear mixtures always have $\beta_{ij} = 0$ once $n$ kernel cells are used up. **In the Appendix**, we provide visualization examples to illustrate this strategy, and a set of ablative experiments to compare it with other alternatives. In the Experiments section, we also validate its effectiveness with basic ablative experiments (see Figure 3 and Table 8).

## 4 EXPERIMENTS

In this section, we conduct comprehensive experiments on image classification, object detection and instance segmentation to evaluate the effectiveness of our KernelWarehouse ("**KW**" for short), compare it with other attention based methods, and provide lots of ablations to study its design.

### 4.1 IMAGE CLASSIFICATION ON IMAGENET

Our main experiments are conducted on ImageNet dataset (Russakovsky et al., 2015), which consists of over 1.2 million training images and 50,000 validation images with 1,000 object categories.

**ConvNet Backbones.** We select backbones from MobileNetV2 (Sandler et al., 2018), ResNet (He et al., 2016) and ConvNeXt (Liu et al., 2022) families for experiments, including both lightweight networks and larger ones. Specifically, we use MobileNetV2 (1.0×), MobileNetV2 (0.5×), ResNet18, ResNet50 and ConvNeXt-Tiny.

**Experimental Setup.** In the experiments, we select DY-Conv (Chen et al., 2020) and ODConv (Li et al., 2022) as our key reference methods, since they are existing top-performing dynamic convolution methods which are also most closely related to KernelWarehouse. We compare our method with them on all the ConvNet backbones except ConvNeXt-Tiny (since there is no publicly available implementation of them on ConvNeXt). To make fair comparisons, all the methods are implemented using the public codes with the same settings for training and testing. In the experiments, we use $b\times$ to denote the convolutional parameter budget of each dynamic convolution method relative to normal convolution, the values of $n$ and $m$ in KernelWarehouse and the experimental details for each network are provided **in the Appendix**.

Table 1: Results comparison on ImageNet with the ResNet18, ResNet50 and ConvNeXt-Tiny backbones. Best results are bolded.

| Models | Params | Top-1 Acc (%) | Top-5 Acc (%) | Models | Params | Top-1 Acc (%) | Top-5 Acc (%) |
|---|---|---|---|---|---|---|---|
| ResNet18 | 11.69M | 70.44 | 89.72 | ResNet50 | 25.56M | 78.44 | 94.24 |
| + DY-Conv (4×) | 45.47M | 73.82 (↑3.38) | 91.48 (↑1.76) | + DY-Conv (4×) | 100.88M | 79.00 (↑0.56) | 94.27 (↑0.03) |
| + ODConv (4×) | 44.90M | 74.45 (↑4.01) | 91.67 (↑1.95) | + ODConv (4×) | 90.67M | 80.62 (↑2.18) | 95.16 (↑0.92) |
| + KW (1/4×) | 4.08M | 72.73 (↑2.29) | 90.83 (↑1.11) | + KW (1/2×) | 17.64M | 79.30 (↑0.86) | 94.71 (↑0.47) |
| + KW (1/2×) | 7.43M | 73.33 (↑2.89) | 91.42 (↑1.70) | + KW (1×) | 28.05M | 80.38 (↑1.94) | 95.19 (↑0.95) |
| + KW (1×) | 11.93M | 74.77 (↑4.33) | 92.13 (↑2.41) | + KW (4×) | 102.02M | **81.05** (↑**2.61**) | **95.21** (↑**0.97**) |
| + KW (2×) | 23.24M | 75.19 (↑4.75) | 92.18 (↑2.46) | ConvNeXt-Tiny | 28.59M | 82.07 | 95.86 |
| + KW (4×) | 45.86M | **76.05** (↑**5.61**) | **92.68** (↑**2.96**) | + KW (1×) | 32.99M | **82.55** (↑**0.48**) | **96.08** (↑**0.22**) |
| | | | | + KW (3/4×) | 24.53M | 82.23 (↑0.16) | 95.88 (↑0.02) |

Table 2: Results comparison on ImageNet with the MobileNetV2 ($1.0\times$, $0.5\times$) backbones trained for 150 epochs. Best results are bolded.

| Models | Params | Top-1 Acc (%) | Top-5 Acc (%) | Models | Params | Top-1 Acc (%) | Top-5 Acc (%) |
|---|---|---|---|---|---|---|---|
| MobileNetV2 ($1.0\times$) | 3.50M | 72.02 | 90.43 | MobileNetV2 ($0.5\times$) | 1.97M | 64.30 | 85.21 |
| + DY-Conv ($4\times$) | 12.40M | 74.94 (↑2.92) | 91.83 (↑1.40) | + DY-Conv ($4\times$) | 4.57M | 69.05 (↑4.75) | 88.37 (↑3.16) |
| + ODConv ($4\times$) | 11.52M | 75.42 (↑3.40) | 92.18 (↑1.75) | + ODConv ($4\times$) | 4.44M | 70.01 (↑5.71) | 89.01 (↑3.80) |
| + KW ($1/2\times$) | 2.65M | 72.59 (↑0.57) | 90.71 (↑0.28) | + KW ($1/2\times$) | 1.47M | 65.19 (↑0.89) | 85.98 (↑0.77) |
| + KW ($1\times$) | 5.17M | 74.68 (↑2.66) | 91.90 (↑1.47) | + KW ($1\times$) | 2.85M | 68.29 (↑3.99) | 87.93 (↑2.72) |
| + KW ($4\times$) | 11.38M | **75.92 (↑3.90)** | **92.22 (↑1.79)** | + KW ($4\times$) | 4.65M | **70.26 (↑5.96)** | **89.19 (↑3.98)** |

Table 3: Results comparison on the MS-COCO 2017 validation set. Best results are bolded.

| Detectors | Backbone Models | Object Detection | | | | | | Instance Segmentation | | | | | |
|---|---|---|---|---|---|---|---|---|---|---|---|---|---|
| | | $AP$ | $AP_{50}$ | $AP_{75}$ | $AP_S$ | $AP_M$ | $AP_L$ | $AP$ | $AP_{50}$ | $AP_{75}$ | $AP_S$ | $AP_M$ | $AP_L$ |
| Mask R-CNN | ResNet50 | 39.6 | 61.6 | 43.3 | 24.4 | 43.7 | 50.0 | 36.4 | 58.7 | 38.6 | 20.4 | 40.4 | 48.4 |
| | + DY-Conv ($4\times$) | 39.6 | 62.1 | 43.1 | 24.7 | 43.3 | 50.5 | 36.6 | 59.1 | 38.6 | 20.9 | 40.2 | 49.1 |
| | + ODConv ($4\times$) | 42.1 | 65.1 | 46.1 | **27.2** | 46.1 | 53.9 | 38.6 | 61.6 | 41.4 | **23.1** | 42.3 | 52.0 |
| | + KW ($1\times$) | 41.8 | 64.5 | 45.9 | 26.6 | 45.5 | 53.0 | 38.4 | 61.4 | 41.2 | 22.2 | 42.0 | 51.6 |
| | + KW ($4\times$) | **42.4** | **65.4** | **46.3** | **27.2** | **46.2** | **54.6** | **38.9** | **62.0** | **41.5** | 22.7 | **42.6** | **53.1** |
| | MobileNetV2 ($1.0\times$) | 33.8 | 55.2 | 35.8 | 19.7 | 36.5 | 44.4 | 31.7 | 52.4 | 33.3 | 16.4 | 34.4 | 43.7 |
| | + DY-Conv ($4\times$) | 37.0 | 58.6 | 40.3 | 21.9 | 40.1 | 47.9 | 34.1 | 55.7 | 36.1 | 18.6 | 37.1 | 46.3 |
| | + ODConv ($4\times$) | 37.2 | 59.4 | 39.9 | 22.6 | 40.0 | 48.0 | 34.5 | 56.4 | 36.3 | 19.3 | 37.3 | 46.8 |
| | + KW ($1\times$) | 36.4 | 58.3 | 39.2 | 22.0 | 39.6 | 47.0 | 33.7 | 55.1 | 35.7 | 18.9 | 36.7 | 45.6 |
| | + KW ($4\times$) | **38.0** | **60.0** | **40.8** | **23.1** | **40.7** | **50.0** | **34.9** | **56.6** | **37.0** | **19.4** | **37.9** | **47.8** |
| | ConvNeXt-Tiny | 43.4 | 65.8 | 47.7 | 27.6 | 46.8 | 55.9 | 39.7 | 62.6 | 42.4 | 23.1 | 43.1 | 53.7 |
| | + KW ($1\times$) | **44.8** | **67.7** | **48.9** | **29.8** | **48.3** | **57.3** | **40.6** | **64.4** | **43.4** | 24.7 | **44.1** | **54.8** |
| | + KW ($3/4\times$) | 44.1 | 66.8 | 48.4 | 29.7 | 47.4 | 56.7 | 40.2 | 63.6 | 43.0 | **24.8** | 43.6 | 54.3 |

**Results Comparison on ResNets and ConvNeXt-Tiny.** In the experiments, we adopt the advanced training strategy recently proposed in ConvNeXt (Liu et al., 2022), with a training schedule of 300 epochs and aggressive augmentations for comparisons on the ResNet18, ResNet50 and ConvNeXt-Tiny. From the results shown in Table 1, we can observe: (1) KW ($4\times$) gets the best results for comparison on the ResNet18, bringing an absolute top-1 gain of 5.61%. Even with 36.45%|65.10% parameter reduction, KW ($1/2\times$)|KW($1/4\times$) brings 2.89%|2.29% top-1 accuracy gain to the ResNet18 baseline; (2) on the larger ResNet50 backbone, while the vanilla dynamic convolution method DY-Conv ($4\times$) shows much lower performance gain, KW ($1/2\times, 1\times, 4\times$) still bring great performance gains to the baseline model. With 30.99% parameter reduction, KW ($1/2\times$) attains a top-1 gain of 0.86% against the baseline model. KW ($4\times$) consistently outperforms both DY-Conv ($4\times$) and ODConv ($4\times$) by 2.05%|0.43% top-1 gain. Beside ResNets, we also apply KernelWarehouse to the ConvNeXt-Tiny backbone to investigate its performance on the state-of-the-art ConvNet architecture. Results show that our method generalizes well on ConvNeXt-Tiny, bringing 0.48%|0.16% top-1 gain to the baseline model with KW ($1\times$)|KW($3/4\times$).

**Results Comparison on MobileNets.** We further apply KernelWarehouse to MobileNetV2 ($1.0\times$, $0.5\times$) to validate its effectiveness on lightweight ConvNet architectures. Since the lightweight MobileNetV2 models have lower capacity compared to ResNet and ConvNeXt models, we don't use aggressive augmentations for MobileNetV2. The results are shown in Table 2. We can see that KernelWarehouse can strike a favorable trade-off between parameter efficiency and representation power for lightweight ConvNets as well as larger ones. Even on the lightweight MobileNetV2 ($1.0\times, 0.5\times$) with 3.50M|1.97M parameters, KW ($1/2\times$) can reduce the model size by 24.29%|25.38% while bringing top-1 gain of 0.57%|0.89%. Similar to the results on the ResNet18 and ResNet50 backbones, KW ($4\times$) also obtains the best results on both MobileNetV2 ($1.0\times$) and MobileNetV2 ($0.5\times$).

## 4.2 Object Detection on MS-COCO

To evaluate the generalization ability of the classification backbone models pre-trained by our method to downstream object detection and instance segmentation tasks, we further conduct comparative experiments on MS-COCO 2017 dataset (Lin et al., 2014), which contains 118,000 training images and 5,000 validation images with 80 object categories.

**Experimental Setup.** We adopt Mask R-CNN (He et al., 2017) as the detection framework, ResNet50 and MobileNetV2 ($1.0\times$) built with different dynamic convolution methods as the backbones which are pre-trained on ImageNet dataset. All the models are trained with standard $1\times$ schedule on MS-COCO dataset. For a fair comparison, we adopt the same settings including data processing pipeline and hyperparameters for all the models. Experimental details are described **in the Appendix**.

**Results Comparison.** The comparison results on Mask R-CNN with different backbones are shown in Table 3. For Mask R-CNN with ResNet50 backbone models, we observe a similar trend to the

Table 4: Ablation of KernelWarehouse with or without kernel partition.

| Models | Kernel Partition | Params | Top-1 Acc (%) | Top-5 Acc (%) |
|---|---|---|---|---|
| ResNet18 | - | 11.69M | 70.44 | 89.72 |
| + KW (1×) | ✓ | 11.93M | **74.77** (↑**4.33**) | **92.13** (↑**2.41**) |
| | - | 11.78M | 70.49 (↑0.05) | 89.84 (↑0.12) |

Table 7: Ablation of KernelWarehouse with or without warehouse sharing between kernels having different dimensions in convolutional blocks.

| Models | Sharing Strategies | Params | Top-1 Acc (%) | Top-5 Acc (%) |
|---|---|---|---|---|
| ResNet50 | - | 25.56M | 78.44 | 94.24 |
| + KW (1×) | With different dimensions | 28.05M | **80.38** (↑**1.94**) | **95.27** (↑**1.03**) |
| | Only with the same dimensions | 26.95M | 79.80 (↑1.36) | 95.01 (↑0.77) |

Table 6: Ablation of KernelWarehouse with warehouse sharing between kernel cells within each stage|within each layer, and without sharing.

| Models | Sharing Strategies | Params | Top-1 Acc (%) | Top-5 Acc (%) |
|---|---|---|---|---|
| ResNet18 | - | 11.69M | 70.44 | 89.72 |
| + KW (1×) | Within each stage | 11.93M | **74.77** (↑**4.33**) | **92.13** (↑**2.41**) |
| | Within each layer | 11.81M | 74.34 (↑3.90) | 91.82 (↑2.10) |
| | Without sharing | 11.78M | 72.49 (↑2.05) | 90.81 (↑1.09) |

Table 8: Ablation of KernelWarehouse with different attention functions.

| Models | Attention Functions | Params | Top-1 Acc (%) | Top-5 Acc (%) |
|---|---|---|---|---|
| ResNet18 | - | 11.69M | 70.44 | 89.72 |
| + KW (1×) | $z_{ij}/\sum_{p=1}^{n}|z_{ip}|$ (ours) | 11.93M | **74.77** (↑**4.33**) | **92.13** (↑**2.41**) |
| | Softmax | 11.93M | 72.67 (↑2.23) | 90.82 (↑1.10) |
| | Sigmoid | 11.93M | 72.09 (↑1.65) | 90.70 (↑0.98) |
| | $max(z_{ij},0)/\sum_{p=1}^{n}|z_{ip}|$ | 11.93M | 72.74 (↑2.30) | 90.86 (↑1.14) |

Table 9: Ablation of KernelWarehouse with or without our attentions initialization strategy.

| Models | Attentions Initialization Strategy | Params | Top-1 Acc (%) | Top-5 Acc (%) |
|---|---|---|---|---|
| ResNet18 | - | 11.69M | 70.44 | 89.72 |
| + KW (1×) | ✓ | 11.93M | **74.77** (↑**4.33**) | **92.13** (↑**2.41**) |
| | - | 11.93M | 73.39 (↑2.95) | 91.24 (↑1.52) |

main experiments on ImageNet dataset: KW (4×) outperforms DY-Conv (4×) and ODConv (4×) on both object detection and instance segmentation tasks. Our KW (1×) brings an AP improvement of 2.2%|2.0% on object detection and instance segmentation tasks, which is on par with ODConv (4×). With the MobileNetV2 (1.0×) backbone, our method yields consistent high accuracy improvements to the baseline, and KW (4×) achieves the best results. With the ConvNeXt-Tiny backbone, the performance gains of KW (1×) and KW (3/4×) to the baseline model become more pronounced on MS-COCO dataset, compared to those on ImageNet dataset, showing higher capacity and good generalization ability of our method to the downstream tasks.

## 4.3 ABLATION STUDIES

For a deeper understanding of KernelWarehouse, we further provide a lot of ablative experiments on ImageNet dataset to study the three key components of KernelWarehouse, namely kernel partition, warehouse sharing and proposed contrasting-driven attention function from different aspects. All the models are trained with the training strategy proposed in ConvNeXt (Liu et al., 2022).

**Effect of Kernel Partition.** Thanks to the kernel partition component, KernelWarehouse can apply denser kernel assembling with a large number of kernel cells. In Table 4, we provide the ablative experiments on the ResNet18 backbone to study the efficacy of kernel partition. We can see that by removing kernel partition, the top-1 gain for KernelWarehouse to the baseline sharply decreases from 4.33% to 0.05%, demonstrating its great importance to our method.

**Warehouse Sharing with Different Ranges.** To validate the effectiveness of the warehouse sharing component, we first perform ablative experiments on the ResNet18 backbone with different ranges of warehouse sharing. From the results shown in Table 6, we can see that when sharing warehouse in wider range, KernelWarehouse brings larger performance improvement to the baseline model. It indicates that explicitly enhancing parameter dependencies within the same convolutional layer and across successive layers both can strengthen the network capacity.

**Warehouse Sharing between Kernels with Different Dimensions.** In the mainstream ConvNet designs, a convolutional block mostly contains several kernels having different dimensions ($k \times k \times c \times f$). We next perform ablative experiments on the ResNet50 backbone to explore the effect of warehouse sharing between kernels having different dimensions in convolutional blocks. Results are summarized in Table 7, showing that warehouse sharing between kernels having different dimensions performs better compared to warehouse sharing only between kernels having the same dimensions. Combining the results in Table 6 and Table 7, we can conclude that enhancing the warehouse sharing between more kernel cells in KernelWarehouse mostly leads to better performance.

**Attention Function.** Recall that KernelWarehouse relies on our proposed contrasting-driven attention function. To explore its role, we also conduct ablative experiments to compare the performance of KernelWarehouse with different attention functions on the ResNet18 backbone. According to the results shown in Table 8, the top-1 accuracy gap between our design $z_{ij}/\sum_{p=1}^{n}|z_{ip}|$ and popular Softmax|Sigmoid reaches 2.10%|2.68%, and our design also outperforms another counterpart

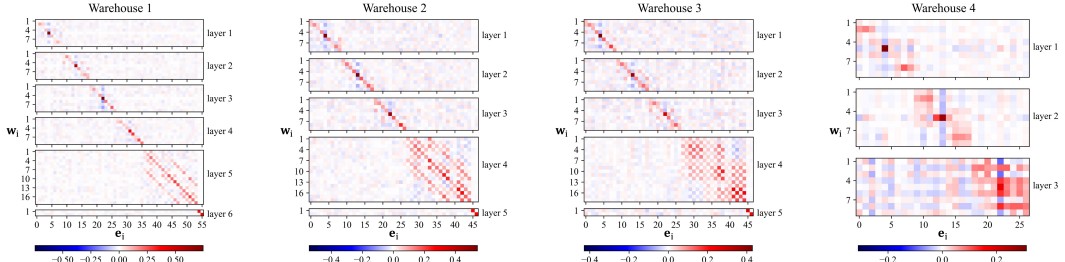

Figure 3: Visualization of statistical mean values of learnt attention $\alpha_{ij}$ in each warehouse. The results are obtained from the pre-trained ResNet18 backbone with KW (1×) for all of the 50,000 images on the ImageNet validation set. Best viewed with zoom-in.

Table 10: Comparison of runtime model speed (frames per second) for different dynamic convolution methods. All the models are tested on an NVIDIA TITAN X GPU (with batch size 100) and a single core of Intel E5-2683 v3 CPU (with batch size 1), separately. The input image size is 224×224.

| Models | Params | Top-1 Acc (%) | Speed on GPU | Speed on CPU |
|---|---|---|---|---|
| ResNet50 | 25.56M | 78.44 | 647.0 | 6.4 |
| + DY-Conv (4×) | 100.88M | 79.00 (↑0.56) | 322.7 | 4.1 |
| + ODConv (4×) | 90.67M | 80.62 (↑2.18) | 142.3 | 2.5 |
| + KW (1/2×) | 17.64M | 79.30 (↑0.86) | 227.8 | 1.5 |
| + KW (1×) | 28.05M | 80.38 (↑1.94) | 265.4 | 1.6 |
| + KW (4×) | 102.02M | 81.05 (↑2.61) | 191.1 | 0.6 |

| Models | Params | Top-1 Acc (%) | Speed on GPU | Speed on CPU |
|---|---|---|---|---|
| MobileNetV2 (1.0×) | 3.50M | 72.02 | 1410.8 | 17.0 |
| + DY-Conv (4×) | 12.40M | 74.94 (↑2.92) | 862.4 | 11.8 |
| + ODConv (4×) | 11.52M | 75.42 (↑3.40) | 536.5 | 11.0 |
| + KW (1/2×) | 2.65M | 72.59 (↑0.57) | 926.0 | 11.6 |
| + KW (1×) | 5.17M | 74.68 (↑2.66) | 798.7 | 10.8 |
| + KW (4×) | 11.38M | 75.92 (↑3.90) | 786.9 | 8.5 |

$max(z_{ij}, 0)/\sum_{p=1}^{n}|z_{ip}|$ by 2.03%, validating the importance of introducing negative attention values in KernelWarehouse to encourage the network to learn adversarial attention relationships.

**Attentions Initialization Strategy.** To help the optimization of KernelWarehouse in the early training stage, $\beta_{ij}$ with temperature $\gamma$ is used for initializing the scalar attentions. In the experiments, we use ResNet18 as the backbone to study the effect of our attentions initialization strategy. As shown in Table 9, a proper initialization strategy for scalar attentions is beneficial for a network to learn relationships between linear mixtures and kernel cells, which leads to 1.38% top-1 improvement to the ResNet18 backbone based on KW (1×).

**Visualization.** To have a better understanding of the warehouse sharing mechanism of KernelWarehouse, we further analyze the statistical mean values of $\alpha_{ij}$ to study its learnt attention values. The results are obtained from the pre-trained ResNet18 backbone with KW (1×). The visualization results are shown in Figure 3, from which we can observe: (1) each linear mixture can learn its own distribution of scalar attentions for different kernel cells; (2) in each warehouse, the maximum value of $\alpha_{ij}$ in each row mostly appears in the diagonal line throughout the whole warehouse. It indicates that our attentions initialization strategy can help KernelWarehouse to build one-to-one relationship between linear mixtures and kernel cells according to our setting of $\beta_{ij}$; (3) compared to linear mixtures in different layers, the attentions $\alpha_{ij}$ with higher absolute values for linear mixtures in the same layer have more overlaps. It indicates that parameter dependencies within the same kernel are stronger than those across successive layers, which can be learned by KernelWarehouse.

**Inference Speed and Others.** Table 10 provides experiments for inference speed analysis, from which we can observe: (1) For relatively large backbones like ResNet50, the runtime model speed of KernelWarehouse is faster than ODConv and is slower than DY-Conv on GPU, but is slower than both ODConv and DY-Conv on CPU (this limitation is mainly due to the dense computation of linear mixtures in KernelWarehouse); (2) For lightweight backbones like MobileNetV2, the runtime model speed of KernelWarehouse, ODConv and DY-Conv is at a similar level both on GPU and CPU. Besides, we believe KernelWarehouse could be applied to more deep and large ConvNets, yet we are unable to explore this constrained by our computational resources.

## 5 CONCLUSION

In this paper, we rethink the design of dynamic convolution and present KernelWarehouse. As a more general form of dynamic convolution, KernelWarehouse can improve the performance of modern ConvNets while enjoying parameter efficiency. Experiments on ImageNet and MS-COCO datasets show its great potential. We hope our work would inspire future research in dynamic convolution.

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

# A APPENDIX

## A.1 TRAINING RESNET18 WITH TRADITIONAL TRAINING STRATEGY.

**Experimental Setup.** Recall that in the experiments section of the main paper, we compare our KernelWarehouse with DY-Conv and ODConv on ResNet and MobileNetV2 models. We adopt the advanced training settings proposed in ConvNeXt (Liu et al., 2022), where all models are trained for 300 epochs. In the experiments, we conduct comparative experiments to explore the performance of KernelWarehouse under traditional training strategy which is adopted by both DY-Conv and ODConv. We make comparisons of KernelWarehouse with various state-of-the-art attention based methods to demonstrate its effectiveness, including: (1) SE (Hu et al., 2018b), CBAM (Woo et al., 2018) and ECA (Wang et al., 2020), which focus on recalibration of feature maps; (2) CGC (Lin et al., 2020) and WeightNet (Ma et al., 2020), which focus on adjusting convolutional weights; (3) DY-Conv (Chen et al., 2020), DCD (Li et al., 2021b) and ODConv (Li et al., 2022), which focus on dynamic convolution.

Table 11: Results comparison on ImageNet with the ResNet18 backbone using the traditional training strategy. Best results are bolded.

| Models | Params | Top-1 Acc (%) | Top-5 Acc (%) |
|---|---|---|---|
| ResNet18 | 11.69M | 70.25 | 89.38 |
| + SE (Hu et al., 2018b) | 11.78M | 70.98 (↑0.73) | 90.03 (↑0.65) |
| + CBAM (Woo et al., 2018) | 11.78M | 71.01 (↑0.76) | 89.85 (↑0.47) |
| + ECA (Wang et al., 2020) | 11.69M | 70.60 (↑0.35) | 89.68 (↑0.30) |
| + CGC (Lin et al., 2020) | 11.69M | 71.60 (↑1.35) | 90.35 (↑0.97) |
| + WeightNet (Ma et al., 2020) | 11.93M | 71.56 (↑1.31) | 90.38 (↑1.00) |
| + DCD (Li et al., 2021b) | 14.70M | 72.33 (↑2.08) | 90.65 (↑1.27) |
| + CondConv (8×) (Yang et al., 2019a) | 81.35M | 71.99 (↑1.74) | 90.27 (↑0.89) |
| + DY-Conv (4×) (Chen et al., 2020) | 45.47M | 72.76 (↑2.51) | 90.79 (↑1.41) |
| + ODConv (4×) (Li et al., 2022) | 44.90M | 73.97 (↑3.72) | 91.35 (↑1.97) |
| + KW (1/2×) | 7.43M | 72.81 (↑2.56) | 90.66 (↑1.28) |
| + KW (1×) | 11.93M | 73.67 (↑3.42) | 91.17 (↑1.79) |
| + KW (2×) | 23.24M | **74.03** (↑**3.78**) | **91.37** (↑**1.99**) |
| + KW (4×) | 45.86M | 73.54 (↑3.29) | 90.94 (↑1.56) |

**Results Comparison on ResNets18 with the Traditional Training Strategy.** We adopt the traditional training strategy adopted by lots of previous studies where models are trained for 100 epochs. The results are shown in Table 11. It can be observed that dynamic convolution methods (CondConv, DY-Conv and ODConv), which introduce obviously more extra parameters, tend to have better performance compared with other methods (SE, CBAM, ECA, CGC, WeightNet and DCD). Note that our KW (1/2×), which has 36.45% parameters less than the baseline, can even outperform all the above attention based methods (except ODConv (4×)) including CondConv (8×) and DY-Conv (4×) which increase the model size to about 6.96|3.89 times. Our KW (2×) achieves the best results, which further surpasses ODConv (4×) by 0.06% top-1 gain with roughly only half of its parameters. However, when we increase KW from 2× to 4×, it shows a decline in top-1 gain from 3.78% to 3.29%. Figure 4 illustrates the training and validation accuracy curves of the ResNet18 models trained with ODConv (4×) and our KW (1×, 2×, 4×). We can see that KW (2×) already gets higher

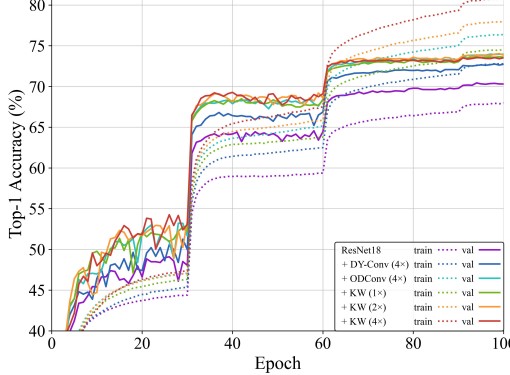

Figure 4: Curves of top-1 training accuracy and validation accuracy of ResNet18 models trained for 100 epochs on ImageNet with DY-Conv (4×), ODConv (4×) and KW (1×, 2×, 4×).

training accuracy than ODConv ($4\times$). While compared to KW ($2\times$), KW ($4\times$) further brings 2.79% improvement on training set but 0.49% drop on validation set. We consider the reason for validation accuracy decline is that KW ($4\times$) largely enhances the capacity of ResNet18 backbone, but also suffers from potential overfitting, when using the traditional training strategy. This overfitting issue can be well resolved by the advanced training strategy adopted in ConvNeXt (Liu et al., 2022) which trains each model with a longer training schedule (300 epochs) and more aggressive augmentations, as can be clearly seen from the results shown in Table 1.

### A.2 Datasets and Implementation Details

#### A.2.1 Image Classification on ImageNet

Recall that we use ResNet (He et al., 2016), MobileNetV2 (Sandler et al., 2018) and ConvNeXt (Liu et al., 2022) families for the main experiments on ImageNet dataset (Russakovsky et al., 2015), which consists of over 1.2 million training images and 50,000 validation images with 1,000 object categories. We use an input image resolution of $224\times224$ for both training and evaluation. All the input images are standardized with mean and standard deviation per channel. For evaluation, we report top-1 and top-5 recognition rates of a single $224\times224$ center crop on the ImageNet validation set. All the experiments are performed on the servers having 8 GPUs. Specifically, the models of ResNet18, MobileNetV2 ($1.0\times$), MobileNetV2 ($0.5\times$) are trained on the servers with 8 NVIDIA Titan X GPUs. The models of ResNet50, ConvNeXt-Tiny are trained on the servers with 8 NVIDIA Tesla V100-SXM3 or A100 GPUs. The training setups for different models are as follows.

**Training setup for ResNet models with the traditional training strategy.** All the models are trained by the stochastic gradient descent (SGD) optimizer for 100 epochs, with a batch size of 256, a momentum of 0.9 and a weight decay of 0.0001. The initial learning rate is set to 0.1 and decayed by a factor of 10 for every 30 epoch. Horizontal flipping and random resized cropping are used for data augmentation. For KernelWarehouse, the temperature $\tau$ linearly reduces from 1 to 0 in the first 10 epochs.

**Training setup for ResNet and ConvNeXt models with the advanced training strategy.** Following the settings of ConvNeXt (Liu et al., 2022), all the models are trained by the AdamW optimizer with $\beta_1 = 0.9, \beta_2 = 0.999$ for 300 epochs, with a batch size of 4096, a momentum of 0.9 and a weight decay of 0.05. The initial learning rate is set to 0.004 and annealed down to zero following a cosine schedule. Randaugment (Cubuk et al., 2020), mixup (Zhang et al., 2018a), cutmix (Yun et al., 2019), random erasing (Zhong et al., 2020) and label smoothing (Szegedy et al., 2016) are used for augmentation. For KernelWarehouse, the temperature $\tau$ linearly reduces from 1 to 0 in the first 20 epochs.

**Training setup for MobileNetV2 models.** All the models are trained by the SGD optimizer for 150 epochs, with a batch size of 256, a momentum of 0.9 and a weight decay of 0.00004. The initial learning rate is set to 0.1 and annealed down to zero following a cosine schedule. Horizontal flipping and random resized cropping are used for data augmentation. For KernelWarehouse, the temperature $\tau$ linearly reduces from 1 to 0 in the first 10 epochs.

#### A.2.2 Object Detection and Instance Segmentation on MS-COCO

Recall that we conduct comparative experiments for object detection and instance segmentation on the MS-COCO 2017 dataset (Lin et al., 2014), which contains 118,000 training images and 5,000 validation images with 80 object categories. We adopt Mask R-CNN as the detection framework, ResNet50 and MobileNetV2 ($1.0\times$) built with different dynamic convolution methods as the backbones which are pre-trained on ImageNet dataset. All the models are trained with a batch size of 16 and standard $1\times$ schedule on the MS-COCO dataset using multi-scale training. The learning rate is decreased by a factor of 10 at the $8^{th}$ and the $11^{th}$ epoch of total 12 epochs. For a fair comparison, we adopt the same settings including data processing pipeline and hyperparameters for all the models. All the experiments are performed on the servers with 8 NVIDIA Tesla V100 GPUs. The attentions initialization strategy is not used for KernelWarehouse during fine-tuning to avoid disrupting the learnt relationships of the pre-trained models between kernel cells and linear mixtures. For evaluation, we report both bounding box Average Precision (AP) and mask AP on the MS-COCO 2017 validation

set, including $AP_{50}$, $AP_{75}$ (AP at different IoU thresholds) and $AP_S$, $AP_M$, $AP_L$ (AP at different scales).

## A.3 VISUALIZATION EXAMPLES OF ATTENTIONS INITIALIZATION STRATEGY

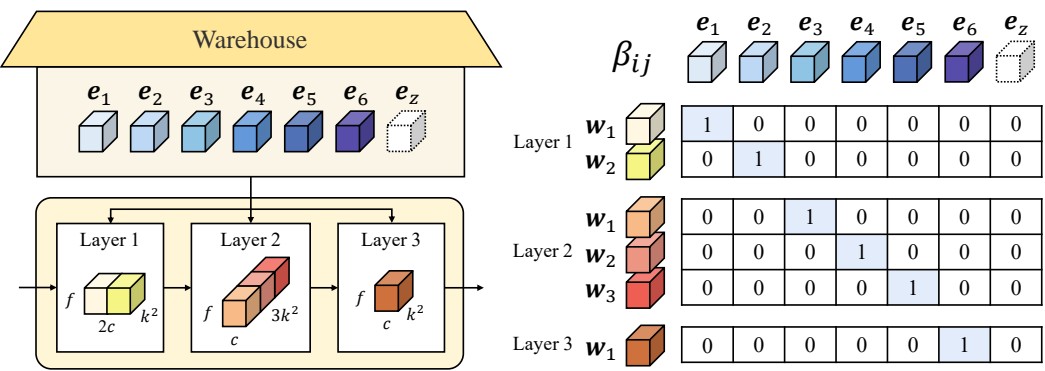

Figure 5: A visualization example of attentions initialization strategy for KW ($1\times$), where both $n$ and $m_t$ equal to 6. It helps the ConvNet to build one-to-one relationships between kernel cells and linear mixtures in the early training stage according to our setting of $\beta_{ij}$. $\mathbf{e}_z$ is a kernel cell that doesn't really exist and it keeps as a zero matrix constantly. In the beginning of the training process when temperature $\tau$ is 1, a ConvNet built with KW ($1\times$) can be roughly seen as a ConvNet with standard convolutions.

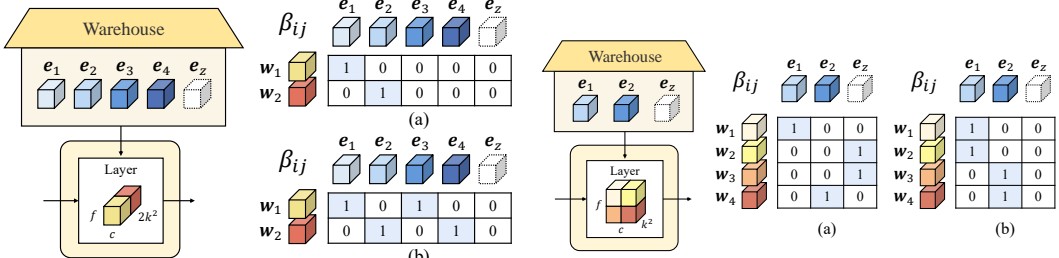

Figure 6: Visualization examples of attentions initialization strategies for KW ($2\times$), where $n = 4$ and $m_t = 2$. (a) our proposed strategy builds one-to-one relationships between kernel cells and linear mixtures; (b) an alternative strategy which builds two-to-one relationships between kernel cells and linear mixtures.

Figure 7: Visualization examples of attentions initialization strategies for KW ($1/2\times$), where $n = 2$ and $m_t = 4$. (a) our proposed strategy builds one-to-one relationships between kernel cells and linear mixtures; (b) an alternative strategy which builds one-to-two relationships between kernel cells and linear mixtures.

Recall that we adopt an attentions initialization strategy for KernelWarehouse using $\tau$ and $\beta_{ij}$. It forces the scalar attentions to be one-hot in the early training stage for building one-to-one relationships between kernel cells and linear mixtures. To give a better understanding of this strategy, we provide visualization examples for KW ($1\times$), KW ($2\times$) and KW ($1/2\times$), respectively. We also provide a set of ablative experiments to compare our proposed strategy with other alternatives.

**Attentions Initialization for KW** ($1\times$). A visualization example of attentions initialization strategy for KW ($1\times$) is shown in Figure 5. In this example, a warehouse $\mathbf{E} = \{\mathbf{e}_1, \ldots, \mathbf{e}_6, \mathbf{e}_z\}$ is shared to 3 neighboring convolutional layers with kernel dimensions of $k \times k \times 2c \times f$, $k \times 3k \times c \times f$ and $k \times k \times c \times f$, respectively. The kernel dimensions are selected for simple illustration. The kernel cells have the same dimensions of $k \times k \times c \times f$. Note that the kernel cell $\mathbf{e}_z$ doesn't really exist and it keeps as a zero matrix constantly. It is only used for attentions normalization but not assembling kernels. This kernel is mainly designed for attentions initialization when $b < 1$ and not counted in the number of kernel cells $n$. In the early training stage, we adopt a strategy to explicitly force every linear mixture to build relationship with one specified kernel cell according to our setting

of $\beta_{ij}$. As shown in Figure 5, we assign one of $\mathbf{e}_1, \ldots, \mathbf{e}_6$ in the warehouse to each of the 6 linear mixtures at the 3 convolutional layers without repetition. So that in the beginning of the training process when temperature $\tau$ is 1, a ConvNet built with KW ($1\times$) can be roughly seen as a ConvNet with standard convolutions. The results of Table 9 in the main manuscript validate the effectiveness of our proposed attentions initialization strategy. Here, we compare it with another alternative. In this alternative strategy, we force every linear mixture to build relationships with all the kernel cells equally by setting all the $\beta_{ij}$ to be 1. The results are shown in Table 12. The all-to-one strategy demonstrates similar performance with KernelWarehouse without using any attentions initialization strategy, while our proposed strategy outperforms it by 1.41% top-1 gain.

Table 12: Ablation of KernelWarehouse with different attentions initialization strategies.

| Models | Attentions Initialization Strategies | Params | Top-1 Acc (%) | Top-5 Acc (%) |
|---|---|---|---|---|
| ResNet18 | - | 11.69M | 70.44 | 89.72 |
| + KW ($1\times$) | 1 kernel cell to 1 linear mixture | 11.93M | **74.77** (↑**4.33**) | **92.13** (↑**2.41**) |
| | all the kernel cells to 1 linear mixture | 11.93M | 73.36 (↑2.92) | 91.41 (↑1.69) |
| | without attentions initialization | 11.93M | 73.39 (↑2.95) | 91.24 (↑1.52) |
| + KW ($4\times$) | 1 kernel cell to 1 linear mixture | 45.86M | **76.05** (↑**5.61**) | **92.68** (↑**2.96**) |
| | 4 kernel cells to 1 linear mixture | 45.86M | 76.03 (↑5.59) | 92.53 (↑2.81) |
| + KW ($1/2\times$) | 1 kernel cell to 1 linear mixture | 7.43M | **73.33** (↑**2.89**) | **91.42** (↑**1.70**) |
| | 1 kernel cell to 2 linear mixtures | 7.43M | 72.89 (↑2.45) | 91.34 (↑1.62) |

**Attentions Initialization for KW ($2\times$).** For KernelWarehouse with $b > 1$, we adopt the same strategy for initializing attentions used in KW ($1\times$). Figure 6(a) provides a visualization example of attentions initialization strategy for KW ($2\times$). For building one-to-one relationships, we assign $\mathbf{e}_1$ to $\mathbf{w}_1$ and $\mathbf{e}_2$ to $\mathbf{w}_2$, respectively. When $b > 1$, another reasonable strategy is to assign multiple kernel cells to every linear mixture without repetition, which is shown in Figure 6(b). We use the ResNet18 backbone based on KW ($4\times$) to compare the two strategies. From the results in Table 12, we can see that our one-to-one strategy performs better.

**Attentions Initialization for KW ($1/2\times$).** For KernelWarehouse with $b < 1$, the number of kernel cells is less than that of linear mixtures, meaning that we cannot adopt the same strategy used for $b \geq 1$. Therefore, we only assign one of the total $n$ kernel cells in the warehouse to $n$ linear mixtures respectively without repetition. And we assign $\mathbf{e}_z$ to all of the remaining linear mixtures. The visualization example for KW ($1/2\times$) is shown in Figure 7(a). When temperature $\tau$ is 1, a ConvNet built with KW ($1/2\times$) can be roughly seen as a ConvNet with group convolutions (groups=2). We also provide comparison results between our proposed strategy and another alternative strategy which assigns one of the $n$ kernel cells to every 2 linear mixtures without repetition. As shown in Table 12, our one-to-one strategy achieves better result again, showing that introducing an extra kernel $\mathbf{e}_z$ for $b < 1$ can help the ConvNet learn more appropriate relationships between kernel cells and linear mixtures. When assigning one kernel cell to multiple linear mixtures, a ConvNet could not balance the relationships between them well.

### A.4 Design Details of KernelWarehouse

In this section, we describe the design details of our KernelWarehouse. The corresponding values of $m$ and $n$ for each of our trained models are provided in the Table 13. Note that the values of $m$ and $n$ are naturally determined according to our setting of the dimensions of the kernel cells, the layers to share warehouses and $b$.

**Design details of Attention Module of KernelWarehouse.** Following existing dynamic convolution methods, KernelWarehouse also adopts a compact SE-typed structure as the attention module $\phi(x)$ (illustrated in Figure 1) to generate attentions for weighting kernel cells in a warehouse. For any convolutional layer with a static kernel $\mathbf{W}$, it starts with a channel-wise global average pooling (GAP) operation that maps the input $\mathbf{x}$ into a feature vector, followed by a fully connected (FC) layer, a rectified linear unit (ReLU), another FC layer, and a contrasting-driven attention function (CAF). The first FC layer reduces the length of the feature vector by 16, and the second FC layer generates $m$ sets of $n$ feature logits in parallel which are finally normalized by our CAF set by set.

**Design details of KernelWarehouse on ResNet18.** Recall that in KernelWarehouse, a warehouse is shared to all same-stage convolutional layers. While the layers are originally divided into different

Table 13: The values of $m$ and $n$ for the ResNet18, ResNet50, ConvNeXt-Tiny, MobileNetV2 (1.0$\times$) and MobileNetV2 (0.5$\times$) backbones based on KernelWarehouse.

| Backbones | $b$ | m | n |
|---|---|---|---|
| ResNet18 | 1/4 | 224, 188, 188, 108 | 56, 47, 47, 27 |
| | 1/2 | 224, 188, 188, 108 | 112, 94, 94, 54 |
| | 1 | 56, 47, 47, 27 | 56, 47, 47, 27 |
| | 2 | 56, 47, 47, 27 | 112, 94, 94, 54 |
| | 4 | 56, 47, 47, 27 | 224, 188, 188, 108 |
| ResNet50 | 1/2 | 348, 416, 552, 188 | 174, 208, 276, 94 |
| | 1 | 87, 104, 138, 47 | 87, 104, 138, 47 |
| | 4 | 87, 104, 138, 47 | 348, 416, 552, 188 |
| ConvNeXt-Tiny | 1 | 16,4,4,4,147,24,147,24,147,24,147,24,147,24,147,24 | 16,4,4,4,147,24,147,24,147,24,147,24,147,24,147,24 |
| | 3/4 | 16,4,4,4,147,24,147,24,147,24,147,24,147,96,147,96 | 16,4,4,4,147,24,147,24,147,24,147,24,147,48,147,48 |
| MobileNetV2 (1.0$\times$) MobileNetV2 (0.5$\times$) | 1/2 | 9, 36, 18, 27, 36, 27, 12, 27, 80, 40 | 9, 36, 18, 27, 36, 27, 6, 27, 40, 20 |
| | 1 | 9, 36, 34, 78, 18, 42, 27, 102, 36, 120, 27, 58, 27 | 9, 36, 34, 78, 18, 42, 27, 102, 36, 120, 27, 58, 27 |
| | 4 | 9, 36, 11, 1, 2, 18, 7, 3, 27, 4, 4, 36, 9, 3, 27, 11, 3, 27, 20 | 36, 144, 44, 4, 8, 72, 28, 12, 108, 16, 16, 144, 36, 12, 108, 44, 12, 108, 80 |

stages according to the resolutions of their input feature maps, the layers are divided into different stages according to their kernel dimensions in our KernelWarehouse. In our implementation, we usually reassign the first layer (or the first two layers) in each stage to the previous stage. An example for the ResNet18 backbone based on KW (1$\times$) is given in Table 14. By reassigning the layers, we can avoid the condition that all the other layers have to be partitioned according to a single layer because of the greatest common dimension divisors. For the ResNet18 backbone, we apply KernelWarehouse to all the convolutional layers except the first one. In each stage, the corresponding warehouse is shared to all of its convolutional layers. For KW (1$\times$), KW (2$\times$) and KW (4$\times$), we use the greatest common dimension divisors for static kernels as the uniform kernel cell dimensions for kernel partition. For KW (1/2$\times$) and KW (1/4$\times$), we use half of the greatest common dimension divisors.

Table 14: The example of warehouse sharing for the ResNet18 backbone based on KW (1$\times$) according to the original stages and reassigned stages.

| Dimensions of Kernel Cells | Original Stages | Layers | Reassigned Stages | Dimensions of Kernel Cells |
|---|---|---|---|---|
| 1$\times$1$\times$64$\times$64 | 1 | 3$\times$3$\times$64$\times$64 | 1 | 1$\times$1$\times$64$\times$64 |
| | | 3$\times$3$\times$64$\times$64 | | |
| | | 3$\times$3$\times$64$\times$64 | | |
| | | 3$\times$3$\times$64$\times$64 | | |
| 1$\times$1$\times$64$\times$128 | 2 | 3$\times$3$\times$64$\times$128 | | |
| | | 3$\times$3$\times$128$\times$128 | 2 | 1$\times$1$\times$128$\times$128 |
| | | 3$\times$3$\times$128$\times$128 | | |
| | | 3$\times$3$\times$128$\times$128 | | |
| 1$\times$1$\times$128$\times$256 | 3 | 3$\times$3$\times$128$\times$256 | | |
| | | 3$\times$3$\times$256$\times$256 | | |
| | | 3$\times$3$\times$256$\times$256 | 3 | 1$\times$1$\times$256$\times$256 |
| | | 3$\times$3$\times$256$\times$256 | | |
| 1$\times$1$\times$256$\times$512 | 4 | 3$\times$3$\times$256$\times$512 | | |
| | | 3$\times$3$\times$512$\times$512 | | |
| | | 3$\times$3$\times$512$\times$512 | 4 | 1$\times$1$\times$512$\times$512 |
| | | 3$\times$3$\times$512$\times$512 | | |

**Design details of KernelWarehouse on ResNet50.** For the ResNet50 backbone, we apply Kernel-Warehouse to all the convolutional layers except the first two layers. In each stage, the corresponding warehouse is shared to all of its convolutional layers. For KW (1$\times$) and KW (4$\times$), we use the greatest common dimension divisors for static kernels as the uniform kernel cell dimensions for kernel partition. For KW (1/2$\times$), we use half of the greatest common dimension divisors.

**Design details of KernelWarehouse on ConvNeXt-Tiny.** For the ConvNeXt backbone, we apply KernelWarehouse to all the convolutional layers. We partition the 9 blocks in the third stage of the ConvNeXt-Tiny backbone into three stages with the equal number of blocks. In each stage, the corresponding three warehouses are shared to the point-wise convolutional layers, the depth-wise convolutional layers and the downsampling layer, respectively. For KW (1$\times$), we use the greatest common dimension divisors for static kernels as the uniform kernel cell dimensions for kernel partition. For KW (3/4$\times$), we apply KW (1/2$\times$) to the point-wise convolutional layers in the last two stages of ConvNeXt backbone using half of the greatest common dimension divisors. And we apply KW (1$\times$) to the other layers using the greatest common dimension divisors.

**Design details of KernelWarehouse on MobileNetV2.** For the MobileNetV2 ($1.0\times$) and MobileNetV2 ($0.5\times$) backbones based on KW ($1\times$) and KW ($4\times$), we apply KernelWarehouse to all the convolutional layers. For MobileNetV2 ($1.0\times$, $0.5\times$) based on KW ($1\times$), the corresponding two warehouses are shared to the point-wise convolutional layers and the depth-wise convolutional layers in each stage, respectively. For MobileNetV2 ($1.0\times$, $0.5\times$) based on KW ($4\times$), the corresponding three warehouses are shared to the depth-wise convolutional layers, the point-wise convolutional layers for channel expansion and the point-wise convolutional layers for channel reduction in each stage, respectively. We use the greatest common dimension divisors for static kernels as the uniform kernel cell dimensions for kernel partition. For the MobileNetV2 ($1.0\times$) and MobileNetV2 ($0.5\times$) backbones based on KW ($1/2\times$), we take the parameters in the attention modules and classifier layer into consideration in order to reduce the total number of parameters. We apply KernelWarehouse to all the depth-wise convolutional layers, the point-wise convolutional layers in the last two stages and the classifier layer. We set $b = 1$ for the point-wise convolutional layers and $b = 1/2$ for the other layers. For the depth-wise convolutional layers, we use the greatest common dimension divisors for static kernels as the uniform kernel cell dimensions for kernel partition. For the point-wise convolutional layers, we use half of the greatest common dimension divisors. For the classifier layer, we use the kernel cell dimensions of $1000\times32$.

## A.5 More Experiments for Studying Other Potentials of KernelWarehouse

In this section, we provide a lot of extra experiments conducted for studying other potentials of KernelWarehouse.

Table 15: Comparison of memory requirements of DY-Conv, ODConv and KernelWarehouse for training and inference. For ResNet50, we set batch size to 128|100 for each gpu during training|inference; for MobileNetV2($1.0\times$), we set batch size to 32|100 for each gpu during training|inference.

| Models | Params | Training Memory (batch size=128) | Inference Memory (batch size=100) | Models | Params | Training Memory (batch size=32) | Inference Memory (batch size=100) |
|---|---|---|---|---|---|---|---|
| ResNet50 | 25.56M | 11,084 MB | 1,249 MB | MobileNetV2 ($1.0\times$) | 3.50M | 2,486 MB | 1,083 MB |
| + DY-Conv ($4\times$) | 100.88M | 24,552 MB | 2,062 MB | + DY-Conv ($4\times$) | 12.40M | 2,924 MB | 1,151 MB |
| + ODConv ($4\times$) | 90.67M | 31,892 MB | 5,405 MB | + ODConv ($4\times$) | 11.52M | 4,212 MB | 1,323 MB |
| + KW ($1/2\times$) | 17.64M | 23,323 MB | 2,121 MB | + KW ($1/2\times$) | 2.65M | 3,002 MB | 1,076 MB |
| + KW ($1\times$) | 28.05M | 23,231 MB | 2,200 MB | + KW ($1\times$) | 5.17M | 2,823 MB | 1,096 MB |
| + KW ($4\times$) | 102.02M | 24,905 MB | 2,762 MB | + KW ($4\times$) | 11.38M | 2,916 MB | 1,144 MB |

**Comparison of Memory Requirements.** From the table 15, we can observe that, for both training and inference, the memory requirements of our method are very similar to those of DY-Conv, and are much smaller than those for ODConv (that generates attention weights along all four dimensions including the input channel number, the output channel number, the spatial kernel size and the kernel number, rather than one single dimension as DY-Conv and KernelWarehouse), showing that our method does not have a potential limitation on memory requirements compared to existing top-performing dynamic convolution methods. The reason is: although KernelWarehouse introduces dense attentive mixturing and assembling operations at the same-stage convolutional layers having a shared warehouse, the memory requirement for these operations is significantly smaller than that for convolutional feature maps and the memory requirement for attention weights are also significantly smaller than that for convolutional weights, under the same convolutional parameter budget $b$.

Table 16: Results comparison on ImageNet with the DeiT-Tiny and DeiT-Small backbones trained for 300 epochs. Best results are bolded.

| Models | Params | Top-1 Acc (%) | Top-5 Acc (%) |
|---|---|---|---|
| DeiT-Tiny | 5.72M | 72.13 | 91.32 |
| + KW ($1\times$) | 6.43M | 73.30 (↑1.17) | 91.46 (↑0.14) |
| + KW ($4\times$) | 21.55M | **76.51 (↑4.38)** | **93.05 (↑1.73)** |
| DeiT-Small | 22.06M | 79.78 | 94.99 |
| + KW ($3/4\times$) | 19.23M | 79.94 (↑0.16) | 95.05 (↑0.06) |
| + KW ($1\times$) | 24.36M | 80.63 (↑0.85) | 95.24 (↑0.25) |
| + KW ($4\times$) | 78.93M | **82.07 (↑2.29)** | **95.70 (↑0.71)** |

**Results Comparison on DeiT.** To validate the effectiveness of KW on vision transformer models, we further perform experiments on DeiT (Touvron et al., 2021) with our KW. In the experiments, we adopt the same settings including data processing pipeline and hyperparameters following DeiT.

For applying KernelWarehouse to a DeiT model, each of split cells of weight matrices for "value and MLP" layers is represented as a linear mixture of kernel warehouse shared across multiple multi-head self-attention blocks and MLP blocks, except the "query" and "key" matrix which are used to compute self-attention. From the results shown in Table 16, it can be seen that: (1) with a small convolutional parameter budget, e.g., $b = 3/4$, KW can get improved model accuracy while reducing model size of DeiT-Small; (2) with a larger convolutional parameter budget, e.g., $b = 4$, KW can significantly improve model accuracy, bringing 4.38%|2.29% absolute top-1 accuracy gain to DeiT-Tiny/DeiT-Small; (3) these performance trends are similar to those reported on ConvNets (see Table 1 and Table 2), demonstrating the appealing generalization ability of our method to different neural network architectures.

Table 17: Ablation of attention functions for different dynamic convolution methods.

| Models | Params | Attention Function | Top-1 Acc (%) | Top-5 Acc (%) |
|---|---|---|---|---|
| ResNet18 | 11.69M | - | 70.44 | 89.72 |
| + DY-Conv (4×) | 45.47M | Softmax | **73.82** | **91.48** |
| | | Ours | 73.74 | 91.45 |
| + ODConv (4×) | 44.90M | Softmax | **74.45** | **91.67** |
| | | Ours | 74.27 | 91.62 |
| + KW (1×) | 11.93M | Softmax | 72.67 | 90.82 |
| | | Ours | **74.77** | **92.13** |
| + KW (4×) | 45.86M | Softmax | 74.31 | 91.75 |
| | | Ours | **76.05** | **92.68** |

**Attention functions for different dynamic convolution methods.** In the table 17, we add experimental results for using the proposed attention function to existing top-performing dynamic convolution methods DY-Conv and ODConv, showing slight drop in model accuracy compared to the original Softmax function. This is because that the proposed method is specialized to fit three unique design properties of KernelWarehouse: (1) the attentive mixture learning is applied to a dense local scale (kernel cells) instead of a holistic kernel scale via kernel partition and warehouse sharing; (2) the number of kernel cells in a warehouse is significantly large (e.g., $n > 100$ instead of $n < 10$); (3) a warehouse is shared to represent every kernel cell in multiple convolutional layers of a ConvNet.

Table 18: Ablation of combining KernelWarehouse with ODConv.

| Models | Params | Top-1 Acc (%) | Top-5 Acc (%) |
|---|---|---|---|
| MobileNetV2 (1.0×) | 3.50M | 72.02 | 90.43 |
| + ODConv (4×) | 11.52M | 75.42 (↑3.40) | 92.18 (↑1.75) |
| + KW (4×) | 11.38M | 75.92 (↑3.90) | 92.22 (↑1.79) |
| + KW & ODConv(4×) | 12.51M | **76.54 (↑4.52)** | **92.35 (↑1.92)** |

**Combining KernelWarehouse with ODConv.** The improvement of KernelWarehouse to ODConv could be further boosted by a simple combination of KernelWarehouse and ODConv to compute attention weights for KernelWarehouse along the aforementioned four dimensions instead of one single dimension. We add experiments to explore this potential, and the results are summarized in the Table 18. We can see that, on the ImageNet dataset with MobileNetV2 (1.0×) backbone, KW & ODConv (4×) brings 1.12% absolute top-1 improvement to ODConv(4×) while retaining the similar model size.

A.6    MORE VISUALIZATION RESULTS FOR LEARNT ATTENTIONS OF KERNELWAREHOUSE

In the main manuscript, we provide visualization results of learnt attention values $\alpha_{ij}$ for the ResNet18 backbone based on KW (1×) (see Figure 3 in the main manuscript). For a better understanding of KernelWarehouse, we provide more visualization results in this section, covering different alternative attention functions, alternative initialization strategies and values of $b$. For all the results, the statistical mean values of learnt attention $\alpha_{ij}$ are obtained using all of the 50,000 images on the ImageNet validation dataset.

**Visualization Results for KernelWarehouse with Different Attention Functions.** The visualization results for KernelWarehouse with different attention functions are shown in Figure 8, which are corresponding to the comparison results of Table 8 in the main manuscript. From which we can observe that: (1) for all of the attention functions, the maximum value of $\alpha_{ij}$ in each row mostly

appears in the diagonal line throughout the whole warehouse. It indicates that our proposed attentions initialization strategy also works for the other three attention functions, which helps our KernelWarehouse to build one-to-one relationships between kernel cells and linear mixtures; (2) with different attention functions, the scalar attentions learnt by KernelWarehouse are obviously different, showing that the attention function plays an important role in our design; (3) compared to the other three functions, the maximum value of $\alpha_{ij}$ in each row tends to be relatively lower for our design (shown in Figure 8(a)). It indicates that the introduction of negative values for scalar attentions can help the ConvNet to enhance warehouse sharing, where each linear mixture not only focuses on the kernel cell assigned to it.

**Visualization Results for KernelWarehouse with Attentions Initialization Strategies.** The visualization results for KernelWarehouse with different attentions initialization strategies are shown in Figure 9, Figure 10 and Figure 11, which are corresponding to the comparison results of Table 12. From which we can observe that: (1) with all-to-one strategy or without initialization strategy, the distribution of scalar attentions learnt by KernelWarehouse seems to be disordered, while our proposed strategy can help the ConvNet learn more appropriate relationships between kernel cells and linear mixtures; (2) for KW ($4\times$) and KW ($1/2\times$), it's hard to directly determine which strategy is better only according to the visualization results. While the results demonstrate that the learnt attentions of KernelWarehouse are highly related to our setting of $\alpha_{ij}$; (3) for KW ($1\times$), KW ($4\times$) and KW ($1/2\times$) with our proposed initialization strategy, some similar patterns of the value distributions can be found. For example, the maximum value of $\alpha_{ij}$ in each row mostly appears in the diagonal line throughout the whole warehouse. It indicates that our proposed strategy can help the ConvNet learn stable relationships between kernel cells and linear mixtures.

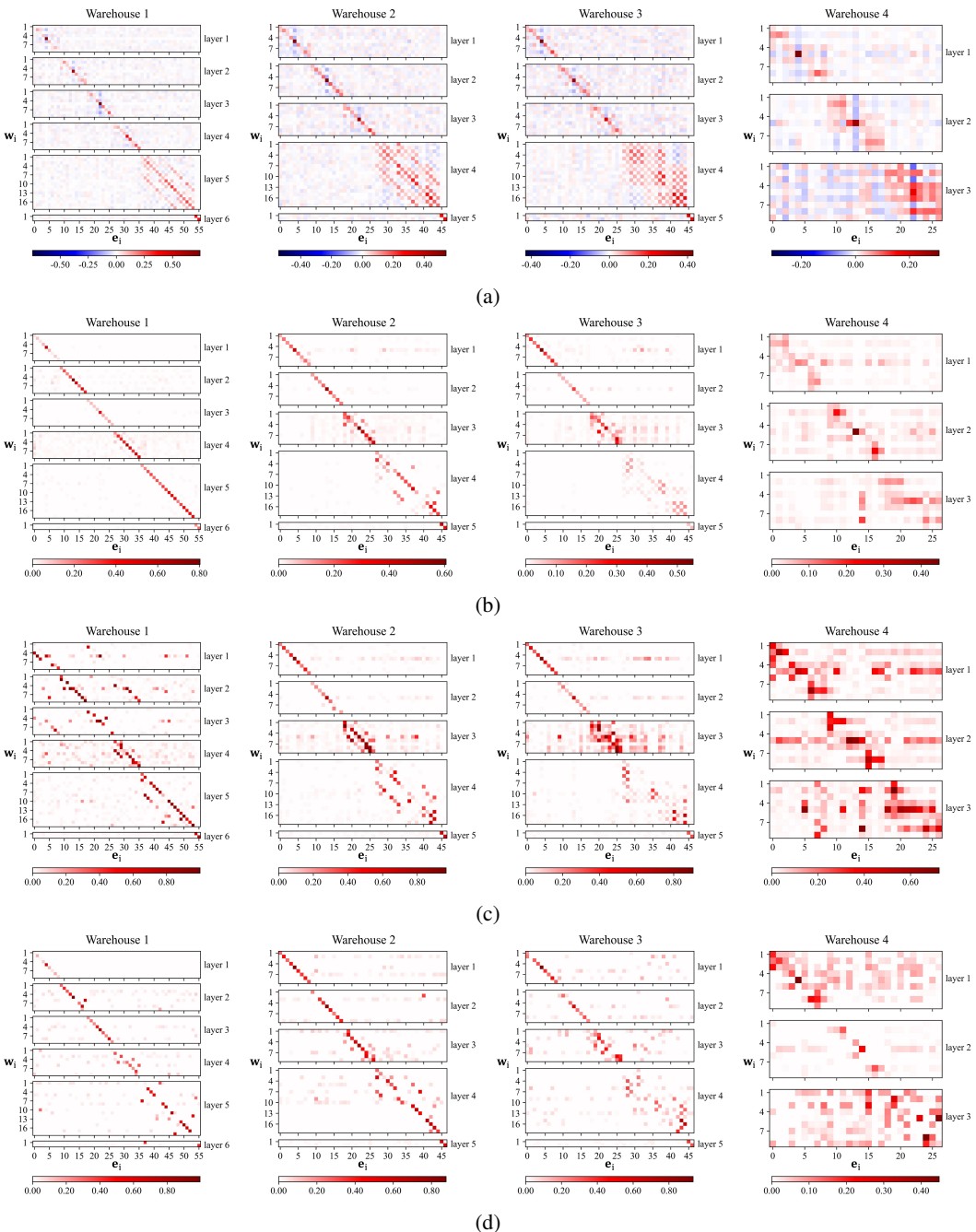

Figure 8: Visualization of statistical mean values of learnt attention $\alpha_{ij}$ in each warehouse for KernelWarehouse with different attention functions. The results are obtained from the pre-trained ResNet18 backbone with KW ($1\times$) for all of the 50,000 images on the ImageNet validation set. Best viewed with zoom-in. The attention functions for the groups of visualization results are as follows: (a) $z_{ij}/\sum_{p=1}^{n}|z_{ip}|$ (our design); (b) softmax; (c) sigmoid; (d) $max(z_{ij},0)/\sum_{p=1}^{n}|z_{ip}|$.

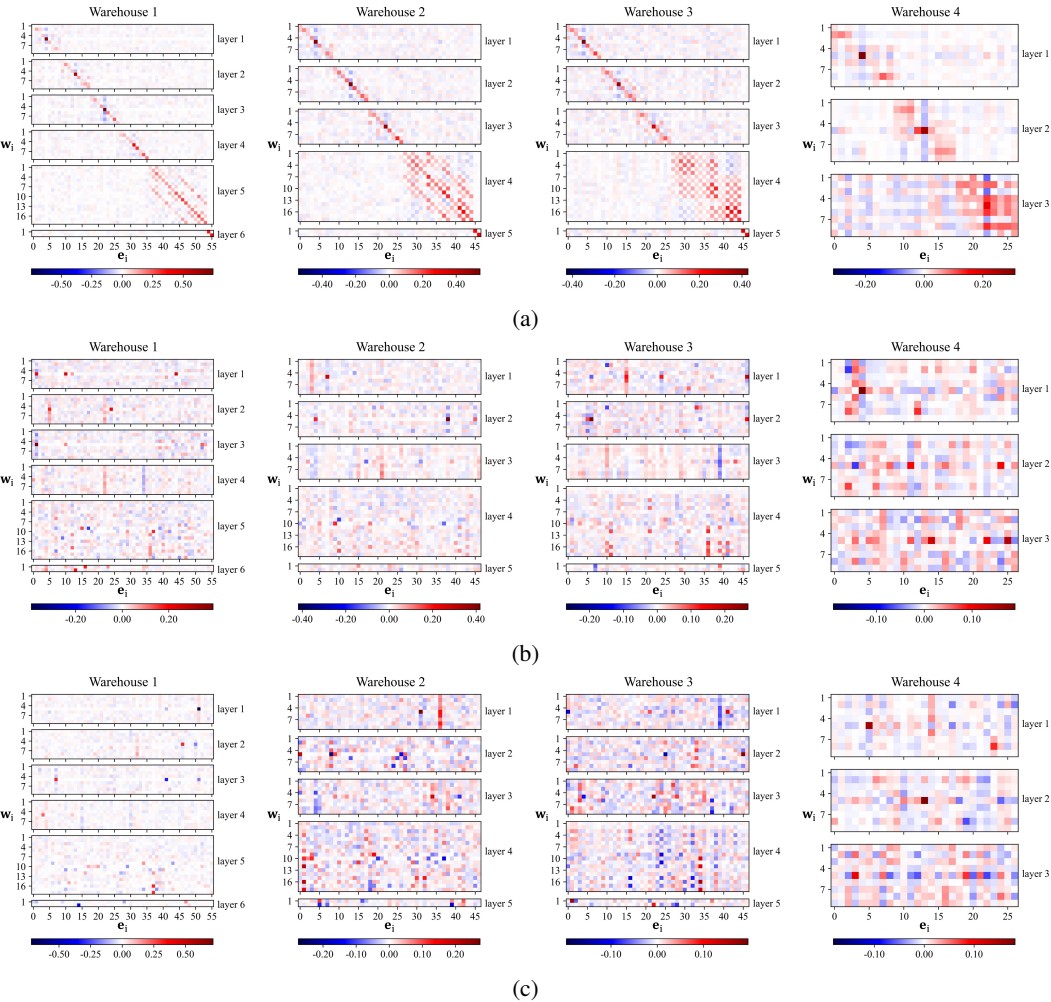

Figure 9: Visualization of statistical mean values of learnt attention $\alpha_{ij}$ in each warehouse for KernelWarehouse with different attentions initialization strategies. The results are obtained from the pre-trained ResNet18 backbone with KW (1×) for all of the 50,000 images on the ImageNet validation set. Best viewed with zoom-in. The attentions initialization strategies for the groups of visualization results are as follows: (a) building one-to-one relationships between kernel cells and linear mixtures; (b) building all-to-one relationships between kernel cells and linear mixtures; (c) without initialization.

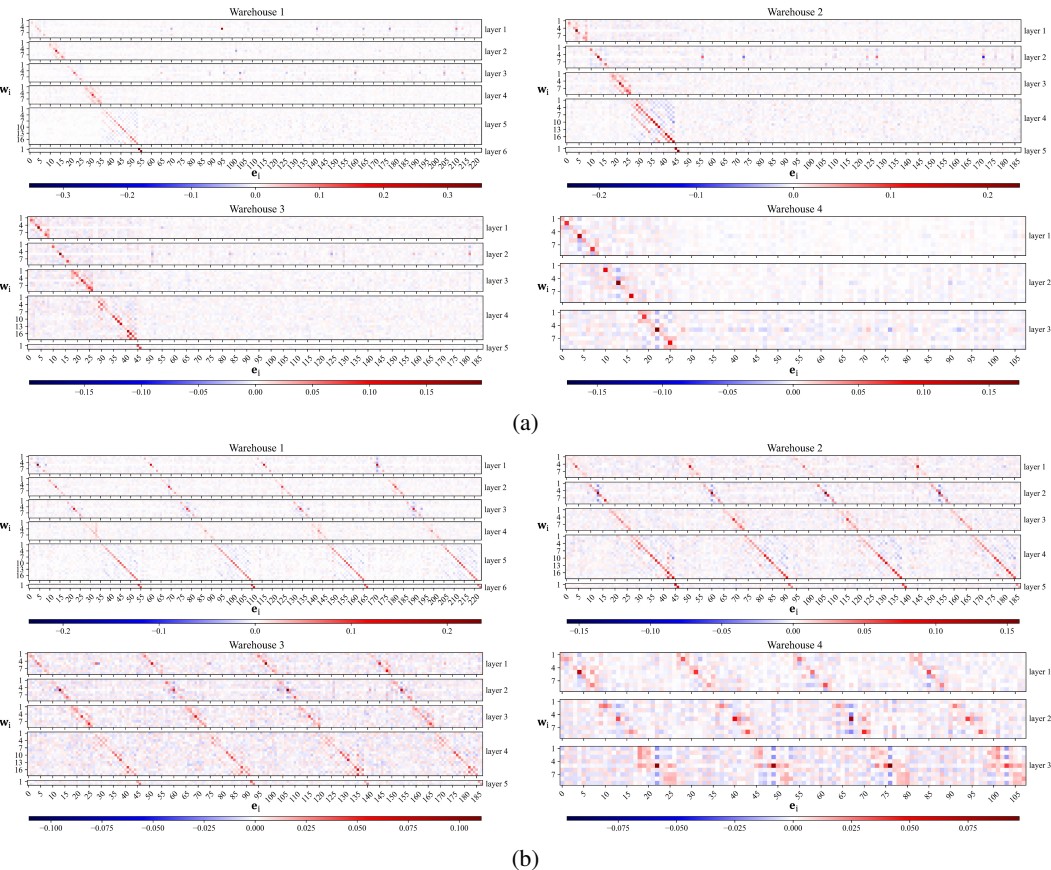

Figure 10: Visualization of statistical mean values of learnt attention $\alpha_{ij}$ in each warehouse for KernelWarehouse with different attentions initialization strategies. The results are obtained from the pre-trained ResNet18 backbone with KW ($4\times$) for all of the 50,000 images on the ImageNet validation set. Best viewed with zoom-in. The attentions initialization strategies for the groups of visualization results are as follows: (a) building one-to-one relationships between kernel cells and linear mixtures; (b) building four-to-one relationships between kernel cells and linear mixtures.

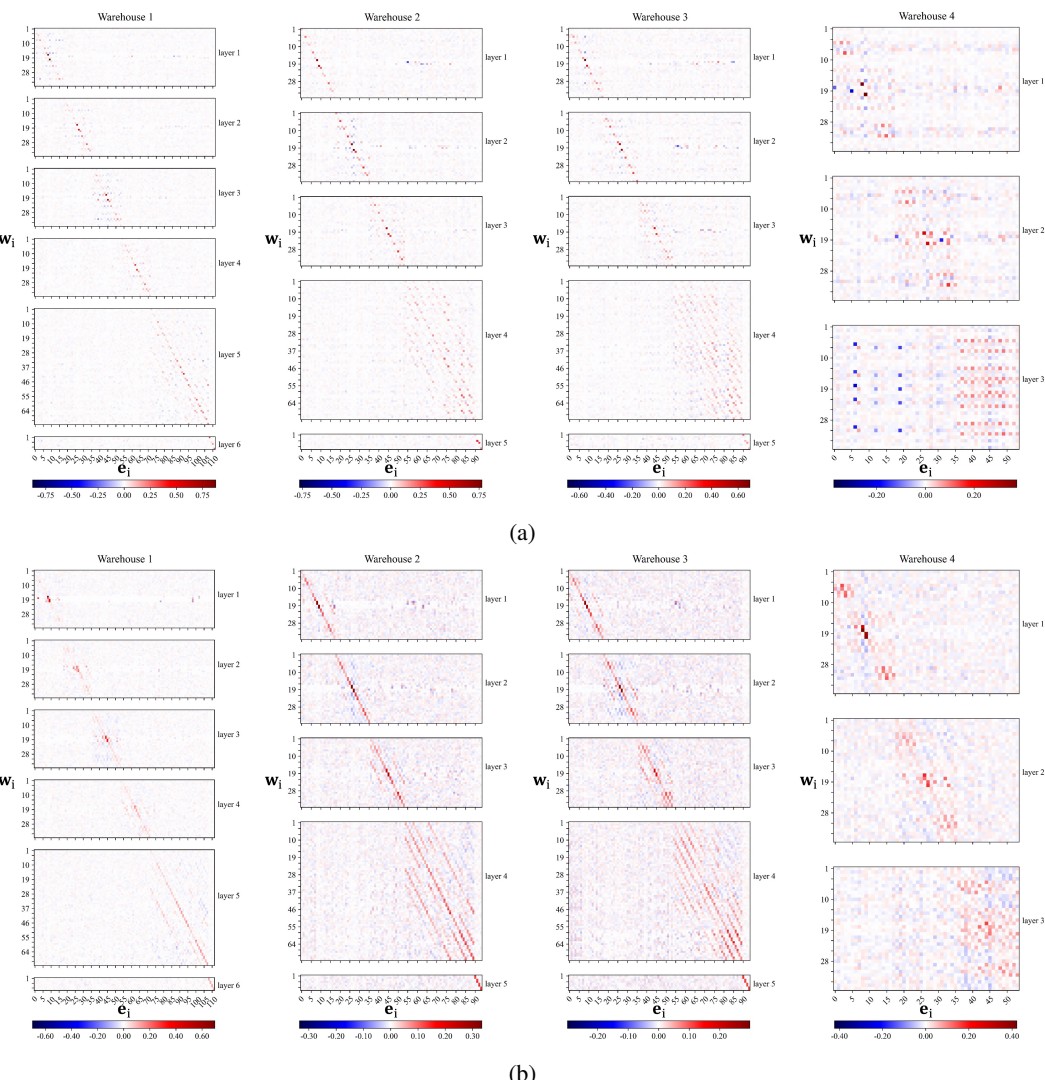

Figure 11: Visualization of statistical mean values of learnt attention $\alpha_{ij}$ in each warehouse for KernelWarehouse with different attentions initialization strategies. The results are obtained from the pre-trained ResNet18 backbone with KW $(1/2\times)$ for all of the 50,000 images on the ImageNet validation set. Best viewed with zoom-in. The attentions initialization strategies for the groups of visualization results are as follows: (a) building one-to-one relationships between kernel cells and linear mixtures; (b) building one-to-two relationships between kernel cells and linear mixtures.

