# OpenReview forum: "KernelWarehouse: Rethinking the Design of Dynamic Convolution"
_ICLR.cc/2024/Conference — Submitted to ICLR 2024_

### Official Review · Reviewer_miVX · 2023-10-18

**Soundness:** 3 good
**Presentation:** 2 fair
**Contribution:** 2 fair
**Rating:** 6
**Confidence:** 2

**Summary:**

In this paper, the authors proposed a new design of dynamic convolution named KernelWarehouse. As a more general form of dynamic convolution, KernelWarehouse can improve the performance of modern ConvNets while enjoying parameter efficiency. Experiments on ImageNet and MS-COCO datasets show its great potential.

**Strengths:**

The proposed method is simple yet effective. It significantly increases the number of dynamic convolutions (>100) while maintaining the efficiency of the network. It is clearly stated how the proposed method is defined and how it works.

**Weaknesses:**

The proposed method claims to achieve the SOTA performance over various vision benchmarks. While it outperforms previous ConvNets, the SOTA models are now mainly transformer-based models. There is no comparison/discussion between the proposed method and transformers.

**Questions:**

In Table 1, why KW 4x is lower than than KW 2x? Any study on this?

---

> ### Author Response · Authors · 2023-11-23
> **Responses to Official Review by Reviewer miVX: Part 1**
>
> Thank you so much for the constructive comments, and the recognition of the novelty and the effectiveness of our proposed method. Please see our below responses to your concerns and questions one by one.
>
> 1.**To your comment regarding the weakness** “The proposed method claims to achieve the SOTA performance over various vision benchmarks. While it outperforms previous ConvNets, the SOTA models are now mainly transformer-based models. There is no comparison/discussion between the proposed method and transformers.”
>
> **Our responses** are: **(1)** We claim that our method achieves SOTA performance over various vision benchmarks **under the condition of** comparing it with existing top-performing dynamic convolution methods such as DY-Conv and ODConv on training the same ConvNet backbones; **(2)** Note dynamic convolution operators are designed to replace normal convolution operators and enhance the capacity of modern ConvNet backbones, and the main motivation of our work is to advance the dynamic convolution research from the perspective of making dynamic convolution operator capable of using a significantly large kernel number ($n>100$ instead of $n<10$) while maintaining parameter efficiency. Therefore, to validate our motivation and method KernelWarehouse, **in experiments we follow popular benchmarking protocols used in dynamic convolution research** and compare our method with existing top-performing dynamic convolution methods on various vision benchmarks by applying them to train different types of ConvNet backbones including lightweight ones and relatively large ones. Under this context, we believe our current experimental comparisons are decent and convincing; **(3)** Yes, the SOTA models are now mainly vision transformer (ViT) models. However, directly comparing ConvNet models trained with our method to ViT models is neither fair nor meaningful due to different architectures and training setups. We think a more convincing way is to validate the generalization ability of our method to ViT backbones. However, because of different feature extraction paradigms (a ConvNet learns local features by applying convolutional layers to image/feature pixels through sliding window strategy and stage-wise down-sampling, while a ViT learns global feature dependencies by adopting a patchify stem where self-attention blocks are applied to non-overlapping same-sized image/feature patches), there is no available research that can apply dynamic convolution to ViTs, to the best of our knowledge. After the paper submission, we further performed pilot experiments on DeiT [1]. Benefiting from kernel partition and warehouse sharing that enable us to apply the attentive mixture learning paradigm to a dense local kernel scale instead of a holistic kernel scale, we find that our method can be well generalized to improve the capacity of seminal ViT architectures by representing each of split cells of weight matrices for “value and MLP” layers as a linear mixture of kernel warehouse shared across multiple multi-head self-attention blocks and MLP blocks, except the “query” and “key” matrix which are already used to compute self-attention. Detailed results are summarized in the below Table. It can be seen that: **(i)** With a small convolutional parameter budget, e.g., $b=3/4\times$, KW can get improved model accuracy while reducing model size of DeiT-Small; **(ii)** With a larger convolutional parameter budget, e.g., $b=4\times$, KW can significantly improve model accuracy, bringing $4.38$%|$2.29$% absolute top-1 accuracy gain to DeiT-Tiny/DeiT-Small; **(iii)** These performance trends are similar to those reported on ConvNets in our original manuscript (see Table 2 and Table 3), demonstrating the appealing generalization ability of our method to different neural network architectures.
>
>
> [1] Hugo Touvron, et al., “Training data-efficient image transformers & distillation through attention”, In ICML,2021.
>
> | Models | Parameters | Top-1 Acc (%) | Top-5 Acc (%) |
> | --- |:---:|:---:|:---:|
> | DeiT-Tiny | 5.72M | 72.13 | 91.32 |
> | + KW (1$\times$) | 6.43M | 73.30 | 91.46 |
> | + KW (4$\times$) | 21.55M | 76.51 | 93.05 |
> | DeiT-Small | 22.06M | 79.78 | 94.99 |
> | + KW (3/4$\times$) | 19.23M | 79.94 | 95.05 |
> | + KW (1$\times$) | 24.36M | 80.63| 95.24 |
> | + KW (4$\times$) | 78.93M | 82.07 | 95.70 |

---

> ### Author Response · Authors · 2023-11-23
> **Responses to Official Review by Reviewer miVX: Part 2**
>
> 2.**To your question** “In Table 1, why KW 4x is lower than KW 2x? Any study on this?”
>
> **Our responses** are: **(1)** Here, the results are reported for the traditional training strategy which trains each model with 100 epochs and simple data augmentations. Under such a training strategy, it tends to incur overfitting on KW when using a larger convolutional parameter budget $b$ like KW($4\times$) against KW($2\times$). Actually, in Figure 2 of the original manuscript, we provided the training and validation accuracy curves of KW($2\times$), KW($4\times$), DY-Conv($4\times$), ODConv($4\times$) and baseline ResNet18 models to clearly illustrate this overfitting issue. It can be seen that KW($4\times$) brings much higher accuracy (2.79% absolute improvement) on training set yet relatively lower accuracy (0.49% absolute drop) on validation set, compared to KW($2\times$) which already attains better performance than DY-Conv($4\times$) and ODConv($4\times$) in terms of both model size and accuracy. That means ResNet18 with KW($4\times$) likely has better model capacity than ResNet18 with KW($2\times$) but may need strong regularization to get final model accuracy improvement on validation set. Such an overfitting issue is the reason why KW($4\times$) is lower than KW($2\times$) in Table 1, and we actually clarified it in our original manuscript (see the paragraph titled **“Results Comparison on ResNets18 with the Traditional Training Strategy”**); **(2)** This overfitting issue can be well resolved by the advanced training strategy adopted in ConvNeXt [2] which trains each model with a longer training schedule (300 epochs) and more aggressive augmentations, as can be clearly seen from the results shown in Table 2. To better explore the potentials of KW and get improved comparisons of KW with existing top-performing dynamic convolution methods DY-Conv($4\times$) and ODConv($4\times$), we adopt this advanced training strategy for our main experiments on relatively large backbones (including ResNets and ConvNeXt) and **get consistently improved model accuracy to all methods** (Table 2 vs. Table 1). Comparatively, our KW always achieves the best results on all backbones, and increasing the convolutional parameter budget $b$ consistently shows improved model accuracy. We also clarified these points in our original manuscript (see the paragraph titled **“Results Comparison on ResNets and ConvNeXt-Tiny with the Advanced Training Strategy”**).
>
> [2] Zhuang Liu, et al., “A convnet for the 2020s”. In CVPR, 2022.
>
> **Finally**, regarding more experiments and discussions that we have made during the rebuttal phase, you are referred to our top-level comments titled **“The Summary of Our Responses to All Official Reviews”**, our responses to the other reviewers, and the revised manuscript.

---

### Official Review · Reviewer_AvKF · 2023-10-29

**Soundness:** 3 good
**Presentation:** 2 fair
**Contribution:** 2 fair
**Rating:** 5
**Confidence:** 3

**Summary:**

This paper proposes a dynamic convolution which can extends the choice of candidates from typical n < 10 to n > 100, while not suffering from the explosion of parameters. The proposed method, KernelWarehouse, can use large kernel numbers when fit a desired parameter budget. This is achieved by exploiting convolutional parameter dependencies within the same layer and across successive layers. Experiments on ImageNet and COCO validate the idea and show significant improvement over baselines.

**Strengths:**

+ Dynamic convolutions, like CondConv, suffer from the explosion of parameters when increasing the kernel choices. This paper proposes a way to alleviate this by exploiting convolutional kernel and layer parameter dependencies. It is a new way to formulate dynamic convolution.

+ Decent improvements are oberseved with different network architectures compared with baselines.

+ Instead of only reporting the parameters, this paper also reports the real runtime in Table 10. Although the proposed method has some limitations, it is still meaningful to report these numbers.

**Weaknesses:**

- The paper writing needs to improve. The introduction only has two major paragraphs and is really hard to parse and digest.

- Although the proposed method could save parameters, it actually requires more time to run (see Table 10), which makes the resulted model far from real deployment.

**Questions:**

- It is counter-intuitive to see KW (4x) performs worse than KW (2x) with additional parameters in Table 1. Could the authors elaborate on this?

---

> ### Author Response · Authors · 2023-11-23
> **Responses to Official Review by Reviewer AvKF: Part 1**
>
> Thank you so much for the constructive comments, and the recognition of the motivation, the novelty and the effectiveness of our proposed method. Please see our below responses to your concerns and questions one by one.
>
> 1.**To your comment regarding the first weakness** “The paper writing needs to improve. The introduction only has two major paragraphs and is really hard to parse and digest.”
>
> **Thanks for your kind suggestion, and our responses** are: **(1)** Indeed, the Introduction section of our original manuscript has two major paragraphs. In paragraph #1, we first describe the concept difference of dynamic convolution to normal convolution by comparing their definitions with a general formulation, then discuss the major limitation of dynamic convolution on parameter efficiency and existing works to address such a limitation, finally point out the current technical gap and our main motivation (namely rethinking dynamic convolution to explore its performance boundary with a significantly large kernel number (e.g., $n > 100$ instead of $n<10$) while enjoying parameter efficiency). In paragraph #2, we first discuss on two critical cues (parameter dependencies within the same layer and parameter dependencies across successive layers) ignored by existing works in dynamic convolution research, which create two technical pathways that serve as the basis to formulate our KernelWarehouse, then describe the key components of KernelWarehouse, namely kernel partition, assembling kernels at multiple convolutional layers with a shared warehouse and a new attention function, to show how KernelWarehouse redefines three basic concepts of “kernels”, “assembling kernels” and “attention function” in dynamic convolution from the perspective of reducing kernel dimension and increasing kernel number significantly. **Although these points are described in a relatively dense but not very detailed manner, they are presented at least in an understandable way, to the best of our understanding**; **(2)** Following your kind suggestion, in our revised manuscript we restructure the Introduction section into four major paragraphs to improve the presentation; **(3)** Furthermore, besides the Figure 1, we add an Additional figure (Figure 2) to better illustrate the key components of KernelWarehouse mentioned above, add an Algorithm table to show the implementation of KernelWarehouse, and give a better clarification of the proposed attention function, following constructive suggestions by Reviewer okim and Reviewer MCn2; **(4)** We believe that the aforementioned revisions could improve the paper writing to a large extent.

---

> ### Author Response · Authors · 2023-11-23
> **Responses to Official Review by Reviewer AvKF: Part 2**
>
> 2.**To your comment regarding the second weakness** “Although the proposed method could save parameters, it actually requires more time to run (see Table 10), which makes the resulted model far from real deployment.”
>
> **Our responses** are: **(1)** According to Table 10, **(a)** for lightweight MobileNetV2 backbone, the runtime model speed of KernelWarehouse is slower than that of DY-Conv **but is faster than** that of existing best-performing ODConv on a single GPU, and their runtime model speeds are at a similar level on a single CPU; **(b)** for relatively large ResNet50 backbone, the runtime model speed of KernelWarehouse is slower than that of DY-Conv **but is still faster than** that of existing best-performing ODConv on a single GPU, and the runtime model speed of KernelWarehouse is obviously lower than that of ODConv|DY-Conv on a single CPU. **More precisely speaking**, the runtime model speed of KernelWarehouse is lower than the vanilla dynamic convolution DY-Conv on both GPU and CPU, **which is the main limitation of our method, as we faithfully clarified in the original manuscript**; **(2)** The speed bottleneck of KernelWarehouse is primarily due to the dense attentive mixture and assembling operations at the same-stage convolutional layers having a shared warehouse. Fortunately, thanks to the parallel property of these operations, the runtime model speed of KernelWarehouse on both GPU and CPU can be improved by the respective optimizations in implementation. Here, we conduct some experiments to explore two direct optimization strategies: **(a)** Firstly, we remove lots of judgement statements used to guarantee the convergence of the model training process which are not necessary at inference phase (but were included in our original benchmarking), by which the runtime model speed of KernelWarehouse on a single GPU can be improved. Detailed results are summarized in the first Table below, showing obvious speed improvements to KernelWarehouse in all cases. For instance, the runtime model speed of KW($4\times$) on a single GPU improves from 567.2|178.5 (frames per second) to 786.9|191.1 for MobileNetV2|ResNet50 backbone, while DY-Conv($4\times$) attains a runtime model speed of 862.4|322.7; **(b)** Secondly, to further accelerate the runtime model speed of KernelWarehouse on both GPU and CPU, we propose an alternative implementation of KernelWarehouse in which we select multiple warehouses with large number of kernels from a ConvNet and split each warehouse shared at the same-stage convolutional layers into two (or more) disjoint same-sized warehouses, and use each warehouse to represent a half of kernels at each convolutional layer. In this way, the number of kernel cells in a warehouse is reduced to half, which can significantly improve the runtime model speed of KernelWarehouse on both GPU and CPU while retaining superior model accuracy compared to existing top-performing dynamic convolution methods DY-Conv and ODConv. Detailed results are summarized in the second Table below, showing obvious speed improvements to KernelWarehouse in all cases. For instance, the runtime model speed of KW($4\times$) on a single GPU improves from 786.9|191.1 (frames per second) to 864.4|282.7 for MobileNetV2|ResNet50 backbone, while DY-Conv($4\times$) attains a runtime model speed of 862.4|322.7; the runtime model speed of KW($4\times$) on a single CPU improves from 8.5|0.6 (frames per second) to 9.7|2.1 for MobileNetV2|ResNet50 backbone, while DY-Conv($4\times$) attains a runtime model speed of 11.8|4.1; **(3)** By these two direct strategies, the runtime model speed of KernelWarehouse on a single GPU|CPU can now better match to that of DY-Conv to a large degree. We will explore more effective strategies to further improve the runtime model speed of our method.

---

> ### Author Response · Authors · 2023-11-23
> **Responses to Official Review by Reviewer AvKF: Part 2-Table**
>
> | Models | Params | Top-1 Acc (%) | Speed on GPU (fps) | Speed on CPU (fps) |
> | --- |:---:|:---:|:---:| :---:|
> | ResNet50 | 25.56M | 78.44 | 647.0 | 6.4 |
> | + DY-Conv ($4\times$) | 100.88M | 79.00 | 322.7 | 4.1 |
> | + ODConv ($4\times$) | 90.67M | 80.62 | 142.3 | 2.5 |
> | + KW ($1/2\times$) | 17.64M | 79.30 | 227.8 (201.0) | 1.5 (1.5) |
> | + KW ($1\times$) | 28.05M | 80.38 | 265.4 (246.1) | 1.6 (1.6) |
> | + KW ($4\times$) | 102.02M | 81.05 | 191.1 (178.5) | 0.6 (0.6) |
> | MobileNetV2 ($1.0\times$ )| 3.50M | 72.02 | 1410.8 | 17.0 |
> | + DY-Conv ($4\times$) | 12.40M | 74.94 | 862.4 | 11.8 |
> | + ODConv ($4\times$) | 11.52M | 75.42 | 536.5 | 11.0 |
> | + KW ($1/2\times$) | 2.65M | 72.59 | 908.3 (825.9) | 11.6 (11.6) |
> | + KW ($1\times$) | 5.17M | 74.68 | 798.7 (575.3) | 10.8 (10.7) |
> | + KW ($4\times$) | 11.38M | 75.92 | 786.9 (567.2) | 8.5 (8.4) |
>
> | Models | Params | Top-1 Acc (%) | Speed on GPU (fps) | Speed on CPU (fps) |
> | --- |:---:|:---:|:---:| :---:|
> | ResNet50 | 25.56M | 78.44 | 647.0 | 6.4 |
> | + DY-Conv ($4\times$) | 100.88M | 79.00 | 322.7 | 4.1 |
> | + ODConv ($4\times$) | 90.67M | 80.62 | 142.3 | 2.5 |
> | + KW ($1/2\times$) | 17.46M (17.64M) | 79.11 | 269.8 (227.8) | 2.3 (1.5) |
> | + KW ($1\times$) | 27.11M (28.05M) | 80.26 | 293.9 (265.4) | 2.6 (1.6) |
> | + KW ($4\times$) | 89.41M (102.02M) | 80.92 | 282.7 (191.1) | 2.1 (0.6) |
> | MobileNetV2 ($1.0\times$ )| 3.50M | 72.02 | 1410.8 | 17.0 |
> | + DY-Conv ($4\times$) | 12.40M | 74.94 | 862.4 | 11.8 |
> | + ODConv ($4\times$) | 11.52M | 75.42 | 536.5 | 11.0 |
> | + KW ($1/2\times$) | 2.57M (2.65M) | 72.53 | 926.0 (908.3) | 11.7 (11.6) |
> | + KW ($1\times$) | 4.89M (5.17M) | 74.51 | 879.2 (798.7) | 11.2 (10.8) |
> | + KW ($4\times$) | 11.16M (11.38M) | 75.83 | 864.4 (786.9) | 9.7 (8.5) |

---

> ### Author Response · Authors · 2023-11-23
> **Responses to Official Review by Reviewer AvKF: Part 3**
>
> 3.**To your question** “It is counter-intuitive to see KW (4x) performs worse than KW (2x) with additional parameters in Table 1. Could the authors elaborate on this?”
>
> **Our responses** are: **(1)** Here, the results are reported for the traditional training strategy which trains each model with 100 epochs and simple data augmentations. Under such a training strategy, it tends to incur overfitting on KW when using a larger convolutional parameter budget $b$ like KW($4\times$) against KW($2\times$). Actually, in Figure 2 of the original manuscript, we provided the training and validation accuracy curves of KW($2\times$), KW($4\times$), DY-Conv($4\times$), ODConv($4\times$) and baseline ResNet18 models to clearly illustrate this overfitting issue. It can be seen that KW($4\times$) brings much higher accuracy (2.79% absolute improvement) on training set yet relatively lower accuracy (0.49% absolute drop) on validation set, compared to KW($2\times$) which already attains better performance than DY-Conv($4\times$) and ODConv($4\times$) in terms of both model size and accuracy. That means ResNet18 with KW($4\times$) likely has better model capacity than ResNet18 with KW($2\times$) but may need strong regularization to get final model accuracy improvement on validation set. Such an overfitting issue is the reason why KW($4\times$) is lower than KW($2\times$) in Table 1, and we actually clarified it in our original manuscript (see the paragraph titled **“Results Comparison on ResNets18 with the Traditional Training Strategy”**); **(2)** This overfitting issue can be well resolved by the advanced training strategy adopted in ConvNeXt [2] which trains each model with a longer training schedule (300 epochs) and more aggressive augmentations, as can be clearly seen from the results shown in Table 2. To better explore the potentials of KW and get improved comparisons of KW with existing top-performing dynamic convolution methods DY-Conv($4\times$) and ODConv($4\times$), we adopt this advanced training strategy for our main experiments on relatively large backbones (including ResNets and ConvNeXt) and **get consistently improved model accuracy to all methods** (Table 2 vs. Table 1). Comparatively, our KW always achieves the best results on all backbones, and increasing the convolutional parameter budget $b$ consistently shows improved model accuracy. We also clarified these points in our original manuscript (see the paragraph titled **“Results Comparison on ResNets and ConvNeXt-Tiny with the Advanced Training Strategy”**).
>
> [2] Zhuang Liu, et al., “A convnet for the 2020s”. In CVPR, 2022.
>
> **Finally**, regarding more experiments and discussions that we have made during the rebuttal phase, you are referred to our top-level comments titled **“The Summary of Our Responses to All Official Reviews”**, our responses to the other reviewers, and the revised manuscript.

---

### Official Review · Reviewer_NHsB · 2023-10-31

**Soundness:** 3 good
**Presentation:** 3 good
**Contribution:** 2 fair
**Rating:** 5
**Confidence:** 5

**Summary:**

This work proposes to improve the previous existing works on dynamic convolution by using kernel partition (dividing the convolution kernel into disjoint parts), sharing parameters across diferent layers of the network and by using a “new attention function”.

**Strengths:**

- the proposed approach is tested on several computer vision tasks and datasets
- the results on improving the parameter efficiency and improving the recognition performance look promising

**Weaknesses:**

Weaknesses*
- The work proposes quite incremental contributions over the dynamic convolution, however, the biggest concern I believe is related to the future use in practice of the proposed version for dynamic convolution. The proposed approach significantly impacts the computational costs in a negative way which can be an important bottleneck for future use in practice. Specifically, as presented in Table 10,  the speed on GPU is reduced from 322.7 to 178.5 images/second (if we compare a similar number of parameters of the model between dynamic convolution and the proposed approach). This is a significant limitation of the approach, which adds even more computational costs (on top of already negative impact of the dynamic convolution over the standard convulsion). Also the memory requirements for training and inference can be a potential limitation, I think the work should also report the memory comparison.

-It looks that a significant gain comes from the  “new attention function” (without this, the work is underperforming the dynamic convolution ), the results are not reported also when using  the “new attention function” in the dynamic convolution framework.

-Not clear why the parameters are decreasing when reducing the sharing levels (Table 6 and 7), I was expecting the opposite.

-The improvements for ConvNeXt look quite marginal, it looks that the approach can have a scalability issue (basically not that useful for larger models)

**Questions:**

See above my concerns

---

> ### Author Response · Authors · 2023-11-23
> **Responses to Official Review by Reviewer NHsB: Part 1**
>
> Thank you so much for the constructive comments, and the recognition of the motivation, the novelty and the effectiveness of our proposed method. Please see our below responses to your concerns and questions one by one.
>
> 1.**To your comment regarding the first weakness** “The work proposes quite incremental contributions over the dynamic convolution, however, the biggest concern I believe is related to the future use in practice of the proposed version for dynamic convolution. The proposed approach significantly impacts the computational costs in a negative way which can be an important bottleneck for future use in practice. Specifically, as presented in Table 10, the speed on GPU is reduced from 322.7 to 178.5 images/second (if we compare a similar number of parameters of the model between dynamic convolution and the proposed approach). This is a significant limitation of the approach, which adds even more computational costs (on top of already negative impact of the dynamic convolution over the standard convulsion). Also the memory requirements for training and inference can be a potential limitation, I think the work should also report the memory comparison.”
>
> **Our responses are in the following three aspects:**
>
> **Firstly, to your biggest concern about the negative impact on the runtime model speed**, **(1)** According to Table 10, the runtime model speed of our method KernelWarehouse is obviously lower than DY-Conv on both GPU and CPU (**but is faster than** existing best-performing ODConv on GPU) under the same convolutional parameter budget $b=4\times$, **which is the main limitation of our method, as we faithfully clarified in the original manuscript**; **(2)** The speed bottleneck of KernelWarehouse is primarily due to the dense attentive mixture and assembling operations at the same-stage convolutional layers having a shared warehouse. Fortunately, thanks to the parallel property of these operations, the runtime model speed of KernelWarehouse on both GPU and CPU can be improved by the respective optimizations in implementation. Here, we conduct some experiments to explore two direct optimization strategies: **(a)** Firstly, we remove lots of judgement statements used to guarantee the convergence of the model training process which are not necessary at inference phase (but were included in our original benchmarking), by which the runtime model speed of KernelWarehouse on a single GPU can be improved. Detailed results are summarized in the first Table below, showing obvious speed improvements to KernelWarehouse in all cases. For instance, the runtime model speed of KW($4\times$) on a single GPU improves from 567.2|178.5 (frames per second) to 786.9|191.1 for MobileNetV2|ResNet50 backbone, while DY-Conv($4\times$) attains a runtime model speed of 862.4|322.7; **(b)** Secondly, to further accelerate the runtime model speed of KernelWarehouse on both GPU and CPU, we propose an alternative implementation of KernelWarehouse in which we select multiple warehouses with large number of kernels from a ConvNet and split each warehouse shared at the same-stage convolutional layers into two (or more) disjoint same-sized warehouses, and use each warehouse to represent a half of kernels at each convolutional layer. In this way, the number of kernel cells in a warehouse is reduced to half, which can significantly improve the runtime model speed of KernelWarehouse on both GPU and CPU while retaining superior model accuracy compared to existing top-performing dynamic convolution methods DY-Conv and ODConv. Detailed results are summarized in the second Table below, showing obvious speed improvements to KernelWarehouse in all cases. For instance, the runtime model speed of KW($4\times$) on a single GPU improves from 786.9|191.1 (frames per second) to 864.4|282.7 for MobileNetV2|ResNet50 backbone, while DY-Conv($4\times$) attains a runtime model speed of 862.4|322.7; the runtime model speed of KW($4\times$) on a single CPU improves from 8.5|0.6 (frames per second) to 9.7|2.1 for MobileNetV2|ResNet50 backbone, while DY-Conv($4\times$) attains a runtime model speed of 11.8|4.1; **(3)** By these two direct strategies, the runtime model speed of KernelWarehouse on a single GPU|CPU can now better match to that of DY-Conv to a large degree. We will explore more effective strategies to further improve the runtime model speed of our method.

---

> ### Author Response · Authors · 2023-11-23
> **Responses to Official Review by Reviewer NHsB: Part 1-Table**
>
> | Models | Params | Top-1 Acc (%) | Speed on GPU (fps) | Speed on CPU (fps) |
> | --- |:---:|:---:|:---:| :---:|
> | ResNet50 | 25.56M | 78.44 | 647.0 | 6.4 |
> | + DY-Conv ($4\times$) | 100.88M | 79.00 | 322.7 | 4.1 |
> | + ODConv ($4\times$) | 90.67M | 80.62 | 142.3 | 2.5 |
> | + KW ($1/2\times$) | 17.64M | 79.30 | 227.8 (201.0) | 1.5 (1.5) |
> | + KW ($1\times$) | 28.05M | 80.38 | 265.4 (246.1) | 1.6 (1.6) |
> | + KW ($4\times$) | 102.02M | 81.05 | 191.1 (178.5) | 0.6 (0.6) |
> | MobileNetV2 ($1.0\times$ )| 3.50M | 72.02 | 1410.8 | 17.0 |
> | + DY-Conv ($4\times$) | 12.40M | 74.94 | 862.4 | 11.8 |
> | + ODConv ($4\times$) | 11.52M | 75.42 | 536.5 | 11.0 |
> | + KW ($1/2\times$) | 2.65M | 72.59 | 908.3 (825.9) | 11.6 (11.6) |
> | + KW ($1\times$) | 5.17M | 74.68 | 798.7 (575.3) | 10.8 (10.7) |
> | + KW ($4\times$) | 11.38M | 75.92 | 786.9 (567.2) | 8.5 (8.4) |
>
> | Models | Params | Top-1 Acc (%) | Speed on GPU (fps) | Speed on CPU (fps) |
> | --- |:---:|:---:|:---:| :---:|
> | ResNet50 | 25.56M | 78.44 | 647.0 | 6.4 |
> | + DY-Conv ($4\times$) | 100.88M | 79.00 | 322.7 | 4.1 |
> | + ODConv ($4\times$) | 90.67M | 80.62 | 142.3 | 2.5 |
> | + KW ($1/2\times$) | 17.46M (17.64M) | 79.11 | 269.8 (227.8) | 2.3 (1.5) |
> | + KW ($1\times$) | 27.11M (28.05M) | 80.26 | 293.9 (265.4) | 2.6 (1.6) |
> | + KW ($4\times$) | 89.41M (102.02M) | 80.92 | 282.7 (191.1) | 2.1 (0.6) |
> | MobileNetV2 ($1.0\times$ )| 3.50M | 72.02 | 1410.8 | 17.0 |
> | + DY-Conv ($4\times$) | 12.40M | 74.94 | 862.4 | 11.8 |
> | + ODConv ($4\times$) | 11.52M | 75.42 | 536.5 | 11.0 |
> | + KW ($1/2\times$) | 2.57M (2.65M) | 72.53 | 926.0 (908.3) | 11.7 (11.6) |
> | + KW ($1\times$) | 4.89M (5.17M) | 74.51 | 879.2 (798.7) | 11.2 (10.8) |
> | + KW ($4\times$) | 11.16M (11.38M) | 75.83 | 864.4 (786.9) | 9.7 (8.5) |

---

> ### Author Response · Authors · 2023-11-23
> **Responses to Official Review by Reviewer NHsB: Part 2**
>
> **Secondly, to your request for the comparison of memory requirements**, we added the results for both training and inference in the below Table. We can observe that, for both training and inference, the memory requirements of our method are very similar to those of DY-Conv, and are much smaller than those for ODConv (that generates attention weights along all four dimensions of the kernel space including the input channel number, the output channel number, the spatial kernel size and the kernel number, rather than one single dimension as DY-Conv and KernelWarehouse), showing that **our method does not have a potential limitation on memory requirements** compared to existing top-performing dynamic convolution methods. The reason is: although KernelWarehouse introduces dense attentive mixturing and assembling operations at the same-stage convolutional layers having a shared warehouse, the memory requirement for these operations is significantly smaller than that for convolutional feature maps and the memory requirement for attention weights are also significantly smaller than that for convolutional weights, under the same convolutional parameter budget $b$.
>
> | Models | Params | Training Memory (batch size=128) | Inference Memory (batch size=100)  |
> | --- |:---:|:---:|:---:|
> | ResNet50 | 25.56M | 11,084MB | 1,249MB |
> | + DY-Conv ($4\times$) | 100.88M | 24,552MB | 2,062MB |
> | + ODConv ($4\times$) | 90.67M | 31,892MB | 5,405MB |
> | + KW ($1/2\times$) | 17.64M | 23,323MB | 2,121MB |
> | + KW ($1\times$) | 28.05M | 23,231MB | 2,200MB |
> | + KW ($4\times$) | 102.02M | 24,905MB | 2,762MB |
>
> | Models | Params | Training Memory (batch size=32) | Inference Memory (batch size=100)  |
> | --- |:---:|:---:|:---:|
> | MobileNetV2 ($1.0\times$)| 3.50M | 2,486MB | 1,083MB |
> | + DY-Conv ($4\times$) | 12.40M | 2,924MB | 1,151MB |
> | + ODConv ($4\times$) | 11.52M | 4,212MB | 1,323MB |
> | + KW ($1/2\times$) | 2.65M | 3,002MB | 1,076MB |
> | + KW ($1\times$) | 5.17M | 2,823MB | 1,096MB |
> | + KW ($4\times$) | 11.38M | 2,916MB | 1,144MB |
>
> **Thirdly, the contributions of our work are three-fold**: **(1)** Our main motivation, rethinking dynamic convolution to explore its performance boundary with a significantly large kernel number (e.g., $n > 100$ instead of $n<10$) while enjoying parameter efficiency, is not explored by existing works; **(2)** two critical insights, namely parameter dependencies within the same layer and parameter dependencies across successive layers, which create two technical pathways that serve as the basis to formulate our method KernelWarehouse, are also not explored by existing works; **(3)** with three key components, namely kernel partition, warehouse sharing to multiple convolutional layers and the proposed attention function, our KernelWarehouse redefines three basic concepts of “kernels”, “assembling kernels” and “attention function” in dynamic convolution from the perspective of reducing kernel dimension and increasing kernel number significantly, and sets performance records for the dynamic convolution on different benchmarks. For the first time, KernelWarehouse can even reduce the model size of a ConvNet while improving the accuracy, as illustrated with MobileNetV2, ResNets and ConvNeXt backbones in Table 2 and Table 3 of the original manuscript; **(4)** Considering the aforementioned points that have been well recognized by other reviewers, we argue that **our contributions are decent rather than quite incremental** over existing dynamic convolution works.

---

> ### Author Response · Authors · 2023-11-23
> **Responses to Official Review by Reviewer NHsB: Part 3**
>
> 2.**To your comment regarding the second weakness** “It looks that a significant gain comes from the “new attention function” (without this, the work is underperforming the dynamic convolution), the results are not reported also when using the “new attention function” in the dynamic convolution framework.”
>
> **Our responses** are: **(1)** Yes, the proposed attention function is essential to our KernelWarehouse, which tightly couples with the other two core components (kernel partition and warehouse sharing) and makes our method can attain the best performance; **(2)** In the below table, we add experimental results for using the proposed attention function to existing top-performing dynamic convolution methods DY-Conv and ODConv, showing slight drop in model accuracy compared to the original Softmax function. This is because that the proposed method is specialized to fit three unique attentive mixture learning properties of KernelWarehouse: **(a)** the attentive mixture learning is applied to a dense local scale (kernel cells) instead of a holistic kernel scale via kernel partition and warehouse sharing; **(b)** the number of kernel cells in a warehouse is significantly large (e.g., $n>100$ instead of $n<10$); **(c)** a warehouse is shared to represent every kernel cell in multiple convolutional layers of a ConvNet.
>
> | Models | Parameters | Attention Function | Top-1 Acc (%) | Top-5 Acc (%)|
> | --- |:---:|:---:|:---:|:---:|
> | ResNet18 | 11.69M | - | 70.44 | 89.72 |
> | + DY-Conv ($4\times$) | 45.47M | Softmax | 73.82 | 91.48 |
> | + DY-Conv ($4\times$) | 45.47M | Ours| 73.74 | 91.45 |
> | + ODConv ($4\times$) | 44.90M | Softmax | 74.45 | 91.67 |
> | + ODConv ($4\times$) | 44.90M | Ours| 74.27 | 91.62 |
> | + KW ($1\times$) | 11.93M | Softmax | 72.67 | 90.82 |
> | + KW ($1\times$) | 11.93M | Ours| 74.77 | 92.13 |
> | + KW ($4\times$) | 45.86M | Softmax | 74.31 | 91.75 |
> | + KW ($4\times$) | 45.86M | Ours| 76.05 | 92.68 |
>
> 3.**To your comment regarding the third weakness** “Not clear why the parameters are decreasing when reducing the sharing levels (Table 6 and 7), I was expecting the opposite.”
>
> **Our responses** are: **(1)** In Table 6 and Table 7, **it is correct** that the model size is decreasing when reducing the sharing levels of warehouse; **(2)** The reason is: As shown in Table 6 and Table 7, all testing cases use the same convolutional parameter budget, $b=1\times$. When reducing the sharing level, the corresponding warehouse will have a reducing number of kernel cells while the size of a kernel cell does not change. As a result, the total attention parameters for linear mixtures are decreasing and the total convolutional parameters ($b$) keep the same, leading to a smaller model size when reducing the sharing level of warehouse.

---

> ### Author Response · Authors · 2023-11-23
> **Responses to Official Review by Reviewer NHsB: Part 4**
>
> 4.**To your comment regarding the last weakness** “The improvements for ConvNeXt look quite marginal, it looks that the approach can have a scalability issue (basically not that useful for larger models)”.
>
> **Our responses** are: **(1)** Note that ConvNeXt [1] adopts an advanced training strategy (with the number of training epochs extended to from the original 100 to 300, lots of aggressive data augmentations such as randaugment, mixup, cutmix and lable smoothing, AdamW optimizer instead of SGD, etc., as we clarified in the original manuscript, see Section A.1.1 of the Appendix for details) that is close to the training strategy used for modern vision transformers. Under such a strong training setup, 0.48%|0.16% top-1 improvement by our KW($1\times$)|KW($3/4\times$) (with 15.7% parameter reduction to the backbone for KW($3/4\times$)) for ConvNeXt pre-trained on ImageNet-1K (Table 2) are obviously smaller compared to the improvements for other backbones, but are acceptable. Furthermore, when transferring the pre-trained ConvNeXt to more challenging downstream tasks (Table 4), the improvements by our KW($1\times$)|KW($3/4\times$) are significantly pronounced, e.g., attaining 1.4%|0.7% mAP gain on object detection; **(2)** Considering that ConvNeXt is a powerful convolutional network architecture designed to match the performance of modern vision transformers, another way to better explore the potentials of our method is to use a larger dataset for model pre-training. Following the paper of ConvNeXt, we conduct a set of new experiments in which ConvNeXt backbone is pre-trained on ImageNet-22K first, then fine-tuned on ImageNet-1K. Detailed results are summarized in below table. We can see that the top-1 improvement of KW($1\times$) is improved from 0.48% to 1.02%, demonstrating the good generalization ability of our method.
>
> | Models | Parameters | Pre-training | Top-1 Acc | Top-5 Acc |
> | --- |:---:|:---:|:---:|:---:|
> | ConvNeXt-Tiny | 28.59M | | 82.07 | 95.86 |
> | ConvNeXt-Tiny | 28.59M | $\checkmark$ | 82.90 | 96.62 |
> | + KW ($1\times$) | 32.99M | | 82.55 | 96.08 |
> | + KW ($1\times$) | 32.99M | $\checkmark$ | 83.92 | 97.03 |
>
> [1] Zhuang Liu, et al., “A convnet for the 2020s”. In CVPR, 2022.
>
> **Finally**, regarding more experiments and discussions that we have made during the rebuttal phase, you are referred to our top-level comments titled **“The Summary of Our Responses to All Official Reviews”**, our responses to the other reviewers, and the revised manuscript.

---

### Official Review · Reviewer_okim · 2023-11-01

**Soundness:** 4 excellent
**Presentation:** 2 fair
**Contribution:** 4 excellent
**Rating:** 8
**Confidence:** 4

**Summary:**

This paper presents KernelWarehouse, a more general form of dynamic convolution that enjoys improved parameter efficiency and expressivity. The paper evaluates the proposed method across several convolutional backbones on MS COCO and ImageNet, consistently leading to improvements over existing approaches. The paper provides an extensive number of ablations corroborating many of the design choices in the proposed method.

**Strengths:**

Altogether I believe this is a very strong submission with some flaws in its presentation. It is a very pleasant read with interesting insights, ablations, visualizations and evaluations. This paper has the potential to have a big impact.

**Weaknesses:**

To my best assessment, this paper does not have any big weaknesses. However, there are a few things regarding the presentation that would improve reading and the clarity of the proposed method.

* The proposed method is simple, yet, in its current form, the paper presents it in a somewhat convoluted manner. I would encourage the authors to restructure the method section (Sec. 3) such that it is presented in an easier way. I want to emphasize that it is not that the paper is not understandable. I just believe that this would help digest the idea in the paper and therefore, probably improve its impact.

  Things that could help:

  > Present the method as an Algorithm.

  > Add an additional figure in which the different components are better illustrated --In image 1, it is not clear how Assemble is done, for example--.

* I strongly encourage the authors to change the name of NAF. The name New Attention Function sounds somewhat odd to me. I would encourage the authors to use a name that describes what the proposed function is doing.

**Questions:**

No questions at this point.

---

> ### Author Response · Authors · 2023-11-23
> **Responses to Official Review by Reviewer okim: Part 1**
>
> Thank you so much for the constructive comments and the recognition of our work.
>
> Following your constructive suggestions, we carefully reconstruct the Method section to improve the writing such that the proposed method KernelWarehouse could be presented in an easier understandable way. Specifically, in the updated manuscript, we have made the following improvements:
>
> **Improvement 1 to the Method section**: Besides Figure 1 showing the overall framework of KernelWarehouse, we add an Additional Figure 2 to better illustrate the different components of KernelWarehouse, showing at the same-stage convolutional layers of a ConvNet backbone how the size of the kernel cell is determined, how kernel partition is performed, how a shared warehouse is constructed, and how kernel assembling is done with the shared warehouse.
>
> **Improvement 2 to the Method section**: We also add an Algorithm table to show how KernelWarehouse is implemented, given a ConvNet backbone and the convolutional parameter budget $b$.
>
> **Improvement 3 to the Method section**: We also change the name of the proposed attention function from “New Attention Function (NAF)” to “Contrasting-driven Attention Function (CAF)”, better describing what our proposed function is doing. Our proposed attention function is specialized to fit three unique attentive mixture learning properties of KernelWarehouse conditioned on kernel partition and warehouse sharing: **(a)** the attentive mixture learning is applied to a dense local scale (kernel cells) instead of a holistic scale (whole kernels); **(b)** the number of kernel cells in a warehouse is significantly large (e.g., $n>100$ instead of $n<10$); **(c)** a warehouse is shared to represent every kernel cell in multiple successive convolutional layers of a ConvNet. Under these design properties, popular attention functions used in existing dynamic convolutional methods such as Sigmoid and Softmax work poorly for KernelWarehouse, but our proposed attention function works well, as tested by experimental results shown in Table 2 and Table 8. The success of the proposed attention function is due to its appealing property that can learn diverse attentions for all linear mixtures simultaneously and make the mixed kernel cells at multiple successive convolutional layers can learn informative and discriminative features hieratically. According to the definition of Equation 3, such a property comes from the binary attention initialization strategy (the second term) and the linear attention normalization function (the first term). Under different convolutional parameter budget $b$, the second term of the proposed attention function ensures that the initial valid kernel cells ($\beta_{ij}=1$) in a shared warehouse are uniformly allocated to represent different linear mixtures at multiple successive convolutional layers at the beginning of the model training, and the first term adopts a linear attention normalization function that **enables the existence of both negative and positive attention weights (popular attention functions do not generate negative attention weights) and encourages the training process to learn contrasting attention relationships** among all linear mixtures sharing the same warehouse. Considering the above motivation and the functioning mechanism, we rename the proposed attention function to “Contrasting-driven Attention Function (CAF)”.

---

> ### Author Response · Authors · 2023-11-23
> **Responses to Official Review by Reviewer okim: Part 2**
>
> **An important improvement to the experiments that you may be interested in**: After the paper submission, we further performed pilot experiments on DeiT [1] to validate the generalization ability of our method to vision transformer (ViT) backbones. Because of different feature extraction paradigms (a ConvNet learns local features by applying convolutional layers to image/feature pixels through sliding window strategy and stage-wise down-sampling, while a ViT learns global feature dependencies by adopting a patchify stem where self-attention blocks are applied to non-overlapping same-sized image/feature patches), there is no available research that can apply dynamic convolution to ViTs, to the best of our knowledge. Benefiting from kernel partition and warehouse sharing that enable us to apply the attentive mixture learning paradigm to a dense local kernel scale instead of a holistic kernel scale, we find that our method can be well generalized to improve the capacity of seminal ViT architectures by representing each of split cells of weight matrices for “value and MLP” layers as a linear mixture of kernel warehouse shared across multiple multi-head self-attention blocks and MLP blocks, except the “query” and “key” matrix which are already used to compute self-attention. Detailed results are summarized in the below Table. It can be seen that: **(i)** With a small convolutional parameter budget, e.g., $b=3/4\times$, KW can get improved model accuracy while reducing model size of DeiT-Small; **(ii)** With a larger convolutional parameter budget, e.g., $b=4\times$, KW can significantly improve model accuracy, bringing $4.38$%|$2.29$% absolute top-1 accuracy gain to DeiT-Tiny/DeiT-Small; **(iii)** These performance trends are similar to those reported on ConvNets in our original manuscript (see Table 2 and Table 3), demonstrating the appealing generalization ability of our method to different neural network architectures.
>
> [1] Hugo Touvron, et al., “Training data-efficient image transformers & distillation through attention”, In ICML,2021.
>
> | Models | Parameters | Top-1 Acc (%)  | Top-5 Acc (%)  |
> | --- |:---:|:---:|:---:|
> | DeiT-Tiny | 5.72M | 72.13 | 91.32 |
> | + KW (1$\times$) | 6.43M | 73.30 | 91.46 |
> | + KW (4$\times$) | 21.55M | 76.51 | 93.05 |
> | DeiT-Small | 22.06M | 79.78 | 94.99 |
> | + KW (3/4$\times$) | 19.23M | 79.94 | 95.05 |
> | + KW (1$\times$) | 24.36M | 80.63| 95.24 |
> | + KW (4$\times$) | 78.93M | 82.07 | 95.70 |
>
> **Finally**, regarding more experiments and discussions that we have made during the rebuttal phase, you are referred to our top-level comments titled **“The Summary of Our Responses to All Official Reviews”**, our responses to the other reviewers, and the revised manuscript.

---

### Official Review · Reviewer_T4nD · 2023-11-07

**Soundness:** 3 good
**Presentation:** 3 good
**Contribution:** 3 good
**Rating:** 6
**Confidence:** 4

**Summary:**

The paper proposes KernelWarehouse as a more general form of dynamic to improve the performance and efficiency of ConvNets. The paper rethinks the basic concept of kernel partition and warehouse. The effectiveness of proposed componets is investigated in detail through the comparison with other attention-based methods in image classification, object detection and instance segmentation.

**Strengths:**

- The paper investigates dynamic convolution in detail with thorough experiments including the comparison with other sota methods in different downstream tasks.
- The ablation of key parameters of the proposed method is given in detail. This paper is written well with organized tables and figures.

**Weaknesses:**

- The improvement of KernelWarehouse is limited as shown in Table1 and Table4 considering two models with the same parameter (+ ODConv (4×) vs + KW (4×)), which can not show the advantage of KernelWarehouse in terms of efficiency and performance.
- The convolutional parameter budget $b$ is an important parameter, how to choose an appropriate parameter of the downstream task. The effect of $b$ on image classification, object detection and instance segmentation is different from the experiment.

**Questions:**

Listed above.

**Details Of Ethics Concerns:**

None.

---

> ### Author Response · Authors · 2023-11-23
> **Responses to Official Review by Reviewer T4nD: Part 1**
>
> Thank you so much for the constructive comments, and the recognition of the motivation, the design and the experimental comparison and the ablation of our work. Please see our below responses to your concerns and questions one by one.
>
> 1.**To your comment regarding the first weakness** “The improvement of KernelWarehouse is limited as shown in Table1 and Table4 considering two models with the same parameter (+ODConv (4×) vs +KW (4×)), which can not show the advantage of KernelWarehouse in terms of efficiency and performance.”
>
> **Our responses** are: **(1)** According to Table 1~Table 4, KW($4\times$) brings an absolute top-1 improvement of 1.60%|0.43%|0.50% to ODConv($4\times$) when pre-training ResNet18|ResNet50|MobileNetV2 backbone on ImageNet-1K dataset, and brings 0.30%-0.80% absolute mAP improvements when transferring the pre-trained ResNet50|MobileNetV2 backbone to downstream object detection and instance segmentation tasks with MS-COCO dataset, while maintaining the similar model size. Although the improvements of KernelWarehouse($4\times$) to ODConv($4\times$) are not as large as the improvements of KernelWarehouse($4\times$) to DY-Conv($4\times$), **these improvements are decent but not limited considering that** ODConv is the existing best-performing dynamic convolution method that generates attention weights along all four dimensions of the kernel space (including the input channel number, the output channel number, the spatial kernel size and the kernel number) rather than one single dimension as DY-Conv and KernelWarehouse; **(2)** **The improvement of KernelWarehouse to ODConv could be further boosted** by a simple combination of KernelWarehouse and ODConv to compute attention weights for KernelWarehouse along the aforementioned four dimensions instead of one single dimension. We add experiments to explore this potential, and the results are summarized in the below Table. We can see that, on ImageNet-1K dataset with MobileNetV2 backbone, KernelWarehouse+ODConv($4\times$) **brings 1.12% absolute top-1 improvement** to ODConv($4\times$) while retaining the similar model size.
>
> | Models | Params | Top-1 Acc (%)  | Top-5 Acc (%) |
> | --- |:---:|:---:|:---:|
> | MobileNetV2 ($1.0\times$) | 3.50M | 72.02 | 90.43 |
> | + ODConv ($4\times$) | 11.52M | 75.42 | 92.18 |
> | + KW ($4\times$) | 11.38M | 75.92 | 92.22 |
> | + KW & ODConv ($4\times$) | 12.51M | 76.54 | 92.35 |

---

> ### Author Response · Authors · 2023-11-23
> **Responses to Official Review by Reviewer T4nD: Part 2**
>
> 2.**To your comment regarding the second weakness** “The convolutional parameter budget $b$ is an important parameter, how to choose an appropriate parameter of the downstream task. The effect of $b$ on image classification, object detection and instance segmentation is different from the experiment.”
>
> **Our responses** are: **(1)** Yes, the convolutional parameter budget $b$ is an important hyper-parameter for our method (and also for existing dynamic convolution methods). When evaluating and comparing the generalization ability of our method and other top-performing dynamic convolution methods on downstream object detection and instance segmentation tasks, we follow the common strategy in the deep learning community to conduct experiments on MS-COCO dataset. Specifically, all dynamic convolution methods are merely used to each ConvNet backbone pre-trained on ImageNet-1K, while the neck and the head of Mask R-CNN still use normal convolution (ensuring that the neck and the head have the same parameters to all dynamic convolution methods for fair comparison). That is, for our method, **the convolutional parameter budget $b$ is defined in terms of a ConvNet backbone but not in terms of the whole Mask R-CNN framework** (consisting of the backbone, the neck and the head); **(2)** In the below Table, we add the results to compare the parameters of the pre-trained ConvNet backbones and the corresponding whole Mask R-CNN models; **(3)** In general, from Table 2, Table 3 and Table 4 in our original manuscript, we can observe that a ConvNet backbone trained by our KernelWarehouse with an increasing convolutional parameter budget $b$ gets consistently better performance on different tasks including image classification, object detection and instance segmentation. Therefore, for downstream tasks, a larger $b$ would be a better choice if there is no constraint on the model size, and otherwise it needs to choose a proper $b$ that can balance the desired model size and accuracy.
>
> | Backbone Models | Total Parameters for Mask R-CNN | Backbone Parameters |
> | --- |:---:|:---:|
> | ResNet50 | 46.45M | 23.51M |
> | + DY-Conv ($4\times$) | 121.77M | 98.83M |
> | + ODConv ($4\times$) | 111.56M | 88.62M |
> | + KW ($1\times$) | 48.94M | 26.00M |
> | + KW ($4\times$) | 122.91M | 99.97M |
> | MobileNetV2 ($1.0\times$)| 23.78M | 2.22M |
> | + DY-Conv ($4\times$) | 32.68M | 11.12M |
> | + ODConv ($4\times$) | 31.80M | 10.24M |
> | + KW ($1\times$) | 25.45M | 3.89M |
> | + KW ($4\times$) | 31.66M | 10.10M |
> | ConvNeXt-Tiny | 48.10M | 27.82M |
> | + KW ($1\times$) | 52.50M | 32.22M |
> | + KW ($3/4\times$) | 44.04M | 23.76M |
>
> **Finally**, regarding more experiments and discussions that we have made during the rebuttal phase, you are referred to our top-level comments titled **“The Summary of Our Responses to All Official Reviews”**, our responses to the other reviewers, and the revised manuscript.

---

### Official Review · Reviewer_MCn2 · 2023-11-07

**Soundness:** 3 good
**Presentation:** 3 good
**Contribution:** 2 fair
**Rating:** 5
**Confidence:** 4

**Summary:**

The paper proposes a novel method for dynamic convolutions, i.e. a method where the convolutional kernels used depend on the input tensor. This idea is typically achieved by having an attention head (e.g. akin to squeeze-and-excitation networks) to provide some linear coefficients that are then used to linearly combine some kernel basis to build the convolutional kernel. The proposed method breaks the kernel into pieces (e.g. across the channel dimension), each piece is constructed in the way described above, and then the pieces are put together to form the kernel. The authors propose some strong sharing methodology in which the kernel basis are shared by the different "kernel pieces" are across layers. The authors also propose a new way of computing the linear coefficients and some initialization strategy that are key to good performance. Experiments show improvements on imagenet, detection and segmentation for a number of convolutional architectures.

**Strengths:**

The paper is clearly motivated and it is easy to understand the differentiation with respect to prior work.

The experimental results are comprehensive, covering several architectures, classification, detection and segmentation, has good ablations and even runtimes. I appreciate for example the inclusion of convnext-tiny and runtime measurements.

The paper is not trivial from a technical standpoint. There seems to be a significant amount of effort and experimentation involved into making the idea work.

Results show high performance and ablations show the need for the different components.

**Weaknesses:**

My main issues are: 1) Architectures have evolved a lot from ResNet18/ResNet50, both in the research as well as the industry areas. 2) Latency is heavily affected. Specifically:

Experiments with ResNet or even MobileNet feel a bit out of sync with the current literature in terms of architecture design. I appreciate the inclusion of convnext-tiny and, while for imagenet results show only moderate gains, there are some clear gains for object detection and segmentation. Besides convolutional architectures, does this kind of technique work for transformers at all?

Latency might be partially due to the lack of optimized CUDA kernels for certain operations, but not completely, as CPU also shows similar issues. I believe the sequential nature of the attention mechanism and the need to put together the kernel are important factors contributing to the latency degradation. These issues are very hard to solve for dynamic convolutions, definitely not just restricted to the current work.

Some of the design choices are key to making the proposed warehouse idea work, e.g. the initialization, temperature and attention design. Do these components work when are combined with other dynamic convolution approaches or is there something in their design that makes them specific to the kernel warehouse idea?

Overall, it feels like the proposed method improves upon prior work on dynamic convolution, but still is not clear in which sense it offers some optimal trade-off.

Comments that might help with the clarity of the paper, no need to reply:
"Kernel partition" could include some information on how the partition is done. I believe it is across the channel dimension but this only becomes clear later.

Tracking the explanations in "Parameter Efficiency and Representation Power" requires a lot of patience and attention. I believe some table with example parameterizations or some exemplification in addition to the current explanations could help the reader a lot.

It would be nice to have some justification for Eq. 3. Right now it looks like the authors came up with it from thin air, but I'm sure there was a motivation or inspiration.

**Questions:**

See comments above

---

> ### Author Response · Authors · 2023-11-23
> **Responses to Official Review by Reviewer MCn2: Part 1**
>
> Thank you so much for the constructive comments, and the recognition of the motivation, the novelty, the experiments and the ablation of our work. Please see our below responses to your concerns and questions one by one.
>
> 1.**To your comment regarding the first main concern on neural network architectures used for performance benchmarking** “1) Architectures have evolved a lot from ResNet18/ResNet50, both in the research as well as the industry areas. Specifically: Experiments with ResNet or even MobileNet feel a bit out of sync with the current literature in terms of architecture design. I appreciate the inclusion of convnext-tiny and, while for imagenet results show only moderate gains, there are some clear gains for object detection and segmentation. Besides convolutional architectures, does this kind of technique work for transformers at all?”
>
> **Our responses** are: **(1)** Dynamic convolution operators are designed to replace normal convolution operators and enhance the capacity of modern ConvNet backbones, and the main motivation of our work is to advance the dynamic convolution research from the perspective of making dynamic convolution operator capable of using a significantly large kernel number ($n>100$ instead of $n<10$) while maintaining parameter efficiency. Therefore, to validate our motivation and proposed method, **in experiments we follow popular benchmarking protocols used in dynamic convolution research** and compare our method with existing top-performing dynamic convolution methods on various vision benchmarks by applying them to train different types of ConvNet backbones including popular lightweight MobileNetV2 and relatively large ResNets, and the recently proposed ConvNeXt (a more powerful ConvNet designed to match the performance of modern vision transformers). Under this context, we believe our current experimental comparisons are decent and convincing; **(2)** Yes, neural network architectures have evolved a lot, and the SOTA models are now mainly vision transformer (ViT) models. However, because of different feature extraction paradigms (a ConvNet learns local features by applying convolutional layers to image/feature pixels through sliding window strategy and stage-wise down-sampling, while a ViT learns global feature dependencies by adopting a patchify stem where self-attention blocks are applied to non-overlapping same-sized image/feature patches), there is no available research that can apply dynamic convolution to ViTs, to the best of our knowledge. After the paper submission, we further performed pilot experiments on DeiT [1]. Benefiting from kernel partition and warehouse sharing that enable us to apply the attentive mixture learning paradigm to a dense local kernel scale instead of a holistic kernel scale, we find that our method can be well generalized to improve the capacity of seminal ViT architectures by representing each of split cells of weight matrices for “value and MLP” layers as a linear mixture of kernel warehouse shared across multiple multi-head self-attention blocks and MLP blocks, except the “query” and “key” matrix which are already used to compute self-attention. Detailed results are summarized in the below Table. It can be seen that: **(i)** With a small convolutional parameter budget, e.g., $b=3/4\times$, KW can get improved model accuracy while reducing model size of DeiT-Small; **(ii)** With a larger convolutional parameter budget, e.g., $b=4\times$, KW can significantly improve model accuracy, bringing $4.38$%|$2.29$% absolute top-1 accuracy gain to DeiT-Tiny/DeiT-Small; **(iii)** These performance trends are similar to those reported on ConvNets in our original manuscript (see Table 2 and Table 3), demonstrating the appealing generalization ability of our method to different neural network architectures.
>
>
> [1] Hugo Touvron, et al., “Training data-efficient image transformers & distillation through attention”, In ICML,2021.
>
> | Models | Parameters | Top-1 Acc (%) | Top-5 Acc (%) |
> | --- |:---:|:---:|:---:|
> | DeiT-Tiny | 5.72M | 72.13 | 91.32 |
> | + KW (1$\times$) | 6.43M | 73.30 | 91.46 |
> | + KW (4$\times$) | 21.55M | 76.51 | 93.05 |
> | DeiT-Small | 22.06M | 79.78 | 94.99 |
> | + KW (3/4$\times$) | 19.23M | 79.94 | 95.05 |
> | + KW (1$\times$) | 24.36M | 80.63| 95.24 |
> | + KW (4$\times$) | 78.93M | 82.07 | 95.70 |

---

> ### Author Response · Authors · 2023-11-23
> **Responses to Official Review by Reviewer MCn2: Part 2**
>
> 2.**To your comment regarding the second main concern on the runtime model speed** “2) Latency is heavily affected. Specifically: Latency might be partially due to the lack of optimized CUDA kernels for certain operations, but not completely, as CPU also shows similar issues. I believe the sequential nature of the attention mechanism and the need to put together the kernel are important factors contributing to the latency degradation. These issues are very hard to solve for dynamic convolutions, definitely not just restricted to the current work.”
>
> **Our responses** are: **(1)** Existing dynamic convolution methods generally suffer from lower runtime model speed compared to normal convolution, just as you said. According to Table 10, the runtime model speed of our method KernelWarehouse is obviously lower than DY-Conv on both GPU and CPU (**but is faster than** existing best-performing ODConv on GPU) under the same convolutional parameter budget $b=4\times$, **which is the main limitation of our method, as we faithfully clarified in the original manuscript**; **(2)** The speed bottleneck of KernelWarehouse is primarily due to the dense attentive mixture and assembling operations at the same-stage convolutional layers having a shared warehouse. Fortunately, thanks to the parallel property of these operations, the runtime model speed of KernelWarehouse on both GPU and CPU can be improved by the respective optimizations in implementation. Here, we conduct some experiments to explore two direct optimization strategies: **(a)** Firstly, we remove lots of judgement statements used to guarantee the convergence of the model training process which are not necessary at inference phase (but were included in our original benchmarking), by which the runtime model speed of KernelWarehouse on a single GPU can be improved. Detailed results are summarized in the first Table below, showing obvious speed improvements to KernelWarehouse in all cases. For instance, the runtime model speed of KW($4\times$) on a single GPU improves from 567.2|178.5 (frames per second) to 786.9|191.1 for MobileNetV2|ResNet50 backbone, while DY-Conv($4\times$) attains a runtime model speed of 862.4|322.7; **(b)** Secondly, to further accelerate the runtime model speed of KernelWarehouse on both GPU and CPU, we propose an alternative implementation of KernelWarehouse in which we select multiple warehouses with large number of kernels from a ConvNet and split each warehouse shared at the same-stage convolutional layers into two (or more) disjoint same-sized warehouses, and use each warehouse to represent a half of kernels at each convolutional layer. In this way, the number of kernel cells in a warehouse is reduced to half, which can significantly improve the runtime model speed of KernelWarehouse on both GPU and CPU while retaining superior model accuracy compared to existing top-performing dynamic convolution methods DY-Conv and ODConv. Detailed results are summarized in the second Table below, showing obvious speed improvements to KernelWarehouse in all cases. For instance, the runtime model speed of KW($4\times$) on a single GPU improves from 786.9|191.1 (frames per second) to 864.4|282.7 for MobileNetV2|ResNet50 backbone, while DY-Conv($4\times$) attains a runtime model speed of 862.4|322.7; the runtime model speed of KW($4\times$) on a single CPU improves from 8.5|0.6 (frames per second) to 9.7|2.1 for MobileNetV2|ResNet50 backbone, while DY-Conv($4\times$) attains a runtime model speed of 11.8|4.1; **(3)** By these two direct strategies, the runtime model speed of KernelWarehouse on a single GPU|CPU can now better match to that of DY-Conv to a large degree. We will explore more effective strategies to further improve the runtime model speed of our method.

---

> ### Author Response · Authors · 2023-11-23
> **Responses to Official Review by Reviewer MCn2: Part 2-Table**
>
> | Models | Params | Top-1 Acc (%) | Speed on GPU (fps) | Speed on CPU (fps) |
> | --- |:---:|:---:|:---:| :---:|
> | ResNet50 | 25.56M | 78.44 | 647.0 | 6.4 |
> | + DY-Conv ($4\times$) | 100.88M | 79.00 | 322.7 | 4.1 |
> | + ODConv ($4\times$) | 90.67M | 80.62 | 142.3 | 2.5 |
> | + KW ($1/2\times$) | 17.64M | 79.30 | 227.8 (201.0) | 1.5 (1.5) |
> | + KW ($1\times$) | 28.05M | 80.38 | 265.4 (246.1) | 1.6 (1.6) |
> | + KW ($4\times$) | 102.02M | 81.05 | 191.1 (178.5) | 0.6 (0.6) |
> | MobileNetV2 ($1.0\times$ )| 3.50M | 72.02 | 1410.8 | 17.0 |
> | + DY-Conv ($4\times$) | 12.40M | 74.94 | 862.4 | 11.8 |
> | + ODConv ($4\times$) | 11.52M | 75.42 | 536.5 | 11.0 |
> | + KW ($1/2\times$) | 2.65M | 72.59 | 908.3 (825.9) | 11.6 (11.6) |
> | + KW ($1\times$) | 5.17M | 74.68 | 798.7 (575.3) | 10.8 (10.7) |
> | + KW ($4\times$) | 11.38M | 75.92 | 786.9 (567.2) | 8.5 (8.4) |
>
> | Models | Params | Top-1 Acc (%) | Speed on GPU (fps) | Speed on CPU (fps) |
> | --- |:---:|:---:|:---:| :---:|
> | ResNet50 | 25.56M | 78.44 | 647.0 | 6.4 |
> | + DY-Conv ($4\times$) | 100.88M | 79.00 | 322.7 | 4.1 |
> | + ODConv ($4\times$) | 90.67M | 80.62 | 142.3 | 2.5 |
> | + KW ($1/2\times$) | 17.46M (17.64M) | 79.11 | 269.8 (227.8) | 2.3 (1.5) |
> | + KW ($1\times$) | 27.11M (28.05M) | 80.26 | 293.9 (265.4) | 2.6 (1.6) |
> | + KW ($4\times$) | 89.41M (102.02M) | 80.92 | 282.7 (191.1) | 2.1 (0.6) |
> | MobileNetV2 ($1.0\times$ )| 3.50M | 72.02 | 1410.8 | 17.0 |
> | + DY-Conv ($4\times$) | 12.40M | 74.94 | 862.4 | 11.8 |
> | + ODConv ($4\times$) | 11.52M | 75.42 | 536.5 | 11.0 |
> | + KW ($1/2\times$) | 2.57M (2.65M) | 72.53 | 926.0 (908.3) | 11.7 (11.6) |
> | + KW ($1\times$) | 4.89M (5.17M) | 74.51 | 879.2 (798.7) | 11.2 (10.8) |
> | + KW ($4\times$) | 11.16M (11.38M) | 75.83 | 864.4 (786.9) | 9.7 (8.5) |

---

> ### Author Response · Authors · 2023-11-23
> **Responses to Official Review by Reviewer MCn2: Part 3**
>
> 3.**To your comments** “Some of the design choices are key to making the proposed warehouse idea work, e.g. the initialization, temperature and attention design. Do these components work when are combined with other dynamic convolution approaches or is there something in their design that makes them specific to the kernel warehouse idea? Overall, it feels like the proposed method improves upon prior work on dynamic convolution, but still is not clear in which sense it offers some optimal trade-off.”
>
> **Our responses** are: **(1)** The proposed attention function, which is defined as Equation 3 consisting of the initialization, the temperature and the generation of the attention, makes our method KernelWarehouse work. Note existing dynamic convolution methods such as DY-Conv and ODConv already use the temperature in their attention functions; **(2)** Combining the proposed attention function with existing dynamic convolution methods leads to slight drops in model accuracy compared to their original attention functions, as can be seen from the experimental results summarized in the first Table below. This is because that the proposed attention function tightly couples with the other two core components (kernel partition and warehouse sharing) and makes our method can attain the best performance by fitting three unique attentive mixture learning properties of KernelWarehouse conditioned on kernel partition and warehouse sharing: **(a)** the attentive mixture learning is applied to a dense local scale (kernel cells) instead of a holistic kernel scale; **(b)** the number of kernel cells in a warehouse is significantly large (e.g., $n\gt 100$ instead of $n\lt 10$); **(c)** a warehouse is shared to represent every kernel cell in multiple convolutional layers of a ConvNet; **(3)** **The performance of KernelWarehouse can be further boosted** by a simple combination of KernelWarehouse and the existing best-performing dynamic convolution method ODConv (that generates attention weights along all four dimensions of the kernel space (including the input channel number, the output channel number, the spatial kernel size and the kernel number) rather than one single dimension as DY-Conv and KernelWarehouse) to compute attention weights for KernelWarehouse along the aforementioned four dimensions instead of one single dimension. We add experiments to explore this potential, and the results are summarized in the second Table below. We can see that, on ImageNet-1K dataset with MobileNetV2 backbone, KernelWarehouse+ODConv($4\times$) **brings 1.12% absolute top-1 improvement** to ODConv($4\times$) while retaining the similar model size; **(4)** **Our work improves prior dynamic convolution research in three-fold**: **(a)** our main motivation, rethinking dynamic convolution to explore its performance boundary with a significantly large kernel number (e.g., $n \gt 100$ instead of $n\lt 10$) while enjoying parameter efficiency, is not explored by existing works; **(b)** two critical insights, namely parameter dependencies within the same layer and parameter dependencies across successive layers, which create two technical pathways that serve as the basis to formulate our method KernelWarehouse, are also not explored by existing works; **(c)** with three interdependent components, namely kernel partition, warehouse sharing to multiple convolutional layers and the proposed attention function, our KernelWarehouse redefines the basic concepts of “kernels”, “assembling kernels” and “attention function” in dynamic convolution from the perspective of reducing kernel dimension and increasing kernel number significantly, and sets performance records for the dynamic convolution on different benchmarks. For the first time, KernelWarehouse can even reduce the model size of a ConvNet while improving the accuracy, as illustrated with MobileNetV2, ResNets and ConvNeXt backbones in Table 2 and Table 3 of the original manuscript.
>
> | Models | Parameters | Attention Function | Top-1 Acc (%) | Top-5 Acc (%) |
> | --- |:---:|:---:|:---:|:---:|
> | ResNet18 | 11.69M | - | 70.44 | 89.72 |
> | + DY-Conv ($4\times$) | 45.47M | Softmax | 73.82 | 91.48 |
> | + DY-Conv ($4\times$) | 45.47M | Ours| 73.74 | 91.45 |
> | + ODConv ($4\times$) | 44.90M | Softmax | 74.45 | 91.67 |
> | + ODConv ($4\times$) | 44.90M | Ours| 74.27 | 91.62 |
> | + KW ($1\times$) | 11.93M | Softmax | 72.67 | 90.82 |
> | + KW ($1\times$) | 11.93M | Ours| 74.77 | 92.13 |
> | + KW ($4\times$) | 45.86M | Softmax | 74.31 | 91.75 |
> | + KW ($4\times$) | 45.86M | Ours| 76.05 | 92.68 |
>
>
> | Models | Params | Top-1 Acc (%) | Top-5 Acc (%) |
> | --- |:---:|:---:|:---:|
> | MobileNetV2 ($1.0\times$) | 3.50M | 72.02 | 90.43 |
> | + ODConv ($4\times$) | 11.52M | 75.42 | 92.18 |
> | + KW ($4\times$) | 11.38M | 75.92 | 92.22 |
> | + KW & ODConv ($4\times$) | 12.51M | 76.54 | 92.35 |

---

> ### Author Response · Authors · 2023-11-23
> **Responses to Official Review by Reviewer MCn2: Part 4**
>
> 4.**To your three constructive suggestions to improve the clarity of the paper** “Comments that might help with the clarity of the paper, no need to reply: "Kernel partition”…I believe…"Parameter Efficiency and Representation Power"…I believe...It would be nice to have some justification for Eq. 3…I'm sure there was a motivation or inspiration”.
>
> **Although you labeled “no need to reply” to these three constructive suggestions as**, we happily follow them to improve the clarity of our paper. Specifically, the updated manuscript includes three-fold presentation improvements:
>
> **Firstly**, “Kernel Partition” is performed across both spatial channel dimensions (illustrated by the “$W$” in Figure 1 and clarified in the paragraph titled Warehouse Sharing in the original manuscript). In the updated manuscript, we add an additional figure (Figure 2) to better illustrate “Kernel Partition” and “Warehouse Construction and Kernel Assembling” components of our method.
>
> **Secondly**, following your constructive suggestion, we use the backbone models pre-trained with KW($1/2\times$)|KW($1\times$)|KW($4\times$) in Table 2 and Table 3 of the original manuscript as exemplifications to better explain how our method can balance "Parameter Efficiency and Representation Power".
>
> **Thirdly**, we also add some justifications to better clarify the motivation and the insight of the proposed attention function (Equation 3). Our proposed attention function is specialized to fit three unique attentive mixture learning properties of KernelWarehouse: **(a)** the attentive mixture learning is applied to a dense local scale (kernel cells) instead of a holistic scale (whole kernels) via kernel partition and warehouse sharing; **(b)** the number of kernel cells in a warehouse is significantly large (e.g., $n>100$ instead of $n<10$); **(c)** a warehouse is shared to represent every kernel cell in multiple successive convolutional layers of a ConvNet. Under these design properties, popular attention functions used in existing dynamic convolutional works such as Sigmoid and Softmax work poorly for KernelWarehouse, but our proposed attention function works well, as tested by experimental results shown in Table 2 and Table 8. The success of the proposed attention function is due to its appealing property that can learn diverse attentions for all linear mixtures simultaneously and make the mixed kernel cells at multiple successive convolutional layers can learn informative and discriminative features hieratically. According to the definition of Equation 3, such a property comes from the binary attention initialization strategy (the second term) and the linear attention normalization function (the first term). Under different convolutional parameter budget $b$, the second term of the proposed attention function ensures that the initial valid kernel cells ($\beta_{ij}=1$) in a shared warehouse are uniformly allocated to represent different linear mixtures at multiple successive convolutional layers at the beginning of the model training, and the first term adopts a linear attention normalization function that **enables the existence of both negative and positive attention weights and encourages the training process to learn contrasting attention relationships** among all linear mixtures sharing the same warehouse facilitated by a temperature. Considering the above motivation and the functioning mechanism, we rename the proposed attention function to “Contrasting-driven Attention Function (CAF)” in the updated manuscript.
>
> **Finally**, regarding more experiments and discussions that we have made during the rebuttal phase, you are referred to our top-level comments titled **“The Summary of Our Responses to All Official Reviews”**, our responses to the other reviewers, and the revised manuscript.

---

### Author Response · Authors · 2023-11-23
**The Summary of Our Responses to All Official Reviews**

Dear Reviewers, Area Chairs, Senior Area Chairs and Program Chairs,

We sincerely thank all six reviewers for their thorough and constructive comments. We are glad that the motivation, the novelty, the experiments, the ablations, and the performance of our work have been widely recognized by reviewers.

In the past about 12 days, we carefully improved the experiments (using all computational resources we have), the clarifications and the discussions of our work to address the concerns, the questions and the requests by all six reviewers. **Summarily, we made the following improvements**:

**(1)** To better explore the potentials and the design of our method KernelWarehouse, we follow the constructive suggestions/requests from Reviewer MCn2, Reviewer T4nD, Reviewer NHsB, Reviewer AvKF and Reviewer miVX, and provide more experiments including: **(a)** a set of experiments shows that our KernelWarehouse can be generalized to significantly improve the performance of seminal vision transformer DeiT while retaining parameter efficiency; **(b)** two sets of experiments show that, under the same convolutional parameter budget, the runtime model speed of our KernelWarehouse is improved to be at a more decent level which can be more closely match to that of the vanilla dynamic convolution DY-Conv on both GPU and CPU by two direct optimization strategies; **(c)** a set of experiments shows that the memory requirements for training and inference of our KernelWarehouse are closely similar to those for the vanilla dynamic convolution method DY-Conv; **(d)** a set of experiments shows that applying the proposed attention function to existing dynamic convolution methods DY-Conv and ODConv leads to slight drops in model accuracy compared to their original attention functions, as it is proposed to fit unique attentive mixture learning properties of KernelWarehouse conditioned on kernel partition and warehouse sharing (clarified in the original manuscript); **(e)** a set of experiments shows that the performance gain of our KernelWarehouse can be further improved when pre-training the ConvNeXt backbone on larger dataset ImageNet-22K instead of ImageNet-1K; **(f)** a set of experiments shows that the performance gain of our KernelWarehouse can be further improved when combining it with ODConv.

**(2)** To have a better presentation of the proposed method, we follow the constructive suggestions by Reviewer okim, Reviewer AvKF and Reviewer MCn2, and improve the writing of the paper by following revisions: **(a)** we carefully reconstruct the Method section, add an Additional figure to better illustrate the different components of our KernelWarehouse, add an Algorithm table to show how our KernelWarehouse is implemented, and improve the clarification of the motivation and the functioning mechanism of the proposed attention function (Equation 3), and use trained backbone models as exemplifications to better explain how our KernelWarehouse can balance parameter efficiency and representation power; **(b)** we also add some  structural modifications to improve the presentation of the Introduction section.

**(3)** We also provide detailed responses to the other concerns/questions/requests raised by each reviewer one by one.

**(4)** In the original Supplementary Material, we already included the code of our KernelWarehouse for image classification, object detection and instance segmentation. We will further update our code based on the above experimental improvements and release it to public. We hope it could help the community to advance dynamic convolution research.

**Finally**, based on the constructive comments by all six reviewers and our responses, **we carefully revised the manuscript of our work**. We hope our detailed responses and the updated manuscript are helpful to address the concerns, the questions and the requests of all six reviewers.

---

### Meta-Review · Area_Chair_MRT8 · 2023-12-05

**Metareview:**

The reviewers are in agreement that the technical contributions in terms of the new type of convolution are interesting; however, it incurs significant additional computational complexity for relatively minor performance gains. Moreover, the derivation of the new convolution is not particularly well-explained or clear from the perspective of algorithm implementation, and has no conceptual basis. I appreciate the significant additional experimentation the authors have conducted during the rebuttal phase, but enhancements of the idea itself in terms of exposition and theory of convolutions is required before it can meet the bar for acceptance.

**Justification For Why Not Higher Score:**

Expositional gaps, lack of conceptual basis, and somewhat cherry picked experiments are not enough to showcase the upsides of the proposed approach relative to prior art.

**Justification For Why Not Lower Score:**

NA

---

### Decision · Program_Chairs · 2024-01-16

Reject